Resource

# Disease- and sex-specific differences in patients with heart valve disease: a proteome study

Sarah Nordmeyer[1,2,3,]*, Milena Kraus[4,]*, Matthias Ziehm[5,6,]*, Marieluise Kirchner[5,6,]*, Marie Schafstedde[1,2,3,6], Marcus Kelm[1,2,6], Sylvia Niquet[5,6], Mariet Mathew Stephen[4], Istvan Baczko[7], Christoph Knosalla[3,8,9], Matthieu-P Schapranow[4], Gunnar Dittmar[10], Michael Gotthardt[3,9,11], Martin Falcke[12], Vera Regitz-Zagrosek[9,13], Titus Kuehne[1,2,3,9], Philipp Mertins[3,5,6]

**Pressure overload in patients with aortic valve stenosis and volume overload in mitral valve regurgitation trigger specific forms of cardiac remodeling; however, little is known about similarities and differences in myocardial proteome regulation. We performed proteome profiling of 75 human left ventricular myocardial biopsies (aortic stenosis = 41, mitral regurgitation = 17, and controls = 17) using high-resolution tandem mass spectrometry next to clinical and hemodynamic parameter acquisition. In patients of both disease groups, proteins related to ECM and cytoskeleton were more abundant, whereas those related to energy metabolism and proteostasis were less abundant compared with controls. In addition, disease group–specific and sex-specific differences have been observed. Male patients with aortic stenosis showed more proteins related to fibrosis and less to energy metabolism, whereas female patients showed strong reduction in proteostasis-related proteins. Clinical imaging was in line with proteomic findings, showing elevation of fibrosis in both patient groups and sex differences. Disease- and sex-specific proteomic profiles provide insight into cardiac remodeling in patients with heart valve disease and might help improve the understanding of molecular mechanisms and the development of individualized treatment strategies.**

## Introduction

Heart valve diseases, such as aortic valve stenosis (AS) and mitral valve regurgitation (MR), are a leading cause of heart failure (HF) (Bouma et al, 1999; Nkomo et al, 2006). Both chronic pressure and volume overload trigger distinctive forms of left ventricular (LV) remodeling with typical concentric hypertrophy in AS and eccentric hypertrophy in MR (Rossi & Carillo, 1991). In an adapted compensated state, patients can remain asymptomatic for years; however, once there is transition into HF and patients become symptomatic, the prognosis is poor if they remain untreated (Kelly et al, 1988; Ryan et al, 2007). In addition, significant sex differences regarding LV remodeling and treatment have been reported (Petrov et al, 2010; Bienjonetti-Boudreau et al, 2021). A more differentiated understanding of the (mal)adaptation in pressure and volume overload including differences between female and male hearts is therefore of high clinical relevance.

At the organ scale, adaptive mechanisms have been extensively studied by clinical imaging techniques to determine phenotypes and ventricular function parameters (Woodard et al, 2006). However, much less is known about cardiac mechanisms at the protein expression level in the human myocardium, mainly because it is challenging to obtain myocardial tissue from human hearts (Doll et al, 2017; Coats et al, 2018; Linscheid et al, 2020). Thus, larger cohorts of patients with AS and MR with comparative proteome analysis of the LV human myocardium also including sex differences have not yet been reported.

The aim of this study was to close this knowledge gap by obtaining in a prospective study detailed proteomic profiles of cardiac remodeling because of severe AS and MR in female and male patients. In addition, we wanted to provide with this study a solid landscape proteome dataset that can be used by the research community for future hypothesis-driven research.

[1]Deutsches Herzzentrum der Charité – Medical Heart Center of Charité and German Heart Institute Berlin, Institute for Cardiovascular Computer-Assisted Medicine, Berlin, Germany    [2]Deutsches Herzzentrum der Charité – Medical Heart Center of Charité and German Heart Institute Berlin, Department of Congenital Heart Disease – Pediatric Cardiology, Berlin, Germany    [3]German Center for Cardiovascular Research (DZHK), Partner Site Berlin, Berlin, Germany    [4]Hasso Plattner Institute for Digital Engineering, Digital Health Center, University of Potsdam, Potsdam, Germany    [5]Max Delbrück Center for Molecular Medicine in the Helmholtz Association, Proteomics Platform, Berlin, Germany    [6]Berlin Institute of Health at Charité – Universitätsmedizin Berlin, Berlin, Germany    [7]Department of Pharmacology and Pharmacotherapy, Interdisciplinary Excellence Centre, University of Szeged, Szeged, Hungary    [8]Deutsches Herzzentrum der Charité – Medical Heart Center of Charité and German Heart, Department of Cardiothoracic and Vascular Surgery, Berlin, Germany    [9]Charité – Universitätsmedizin Berlin, Corporate Member of Freie Universität Berlin and Humboldt-Universität zu Berlin, Berlin, Germany    [10]Proteomics of Cellular Signaling, Luxembourg Institute of Health, Strassen, Luxembourg    [11]Max Delbrück Center for Molecular Medicine in the Helmholtz Association, Neuromuscular and Cardiovascular Cell Biology, Berlin, Germany    [12]Max Delbrück Center for Molecular Medicine, Mathematical Cell Physiology, Berlin, Germany    [13]Department of Cardiology, University Hospital Zürich, University of Zürich, Zürich, Switzerland

Correspondence: snordmeyer@dhzb.de; titus.kuehne@dhzb.de; philipp.mertins@mdc-berlin.de
*Sarah Nordmeyer, Milena Kraus, Matthias Ziehm, and Marieluise Kirchner contributed equally to this work

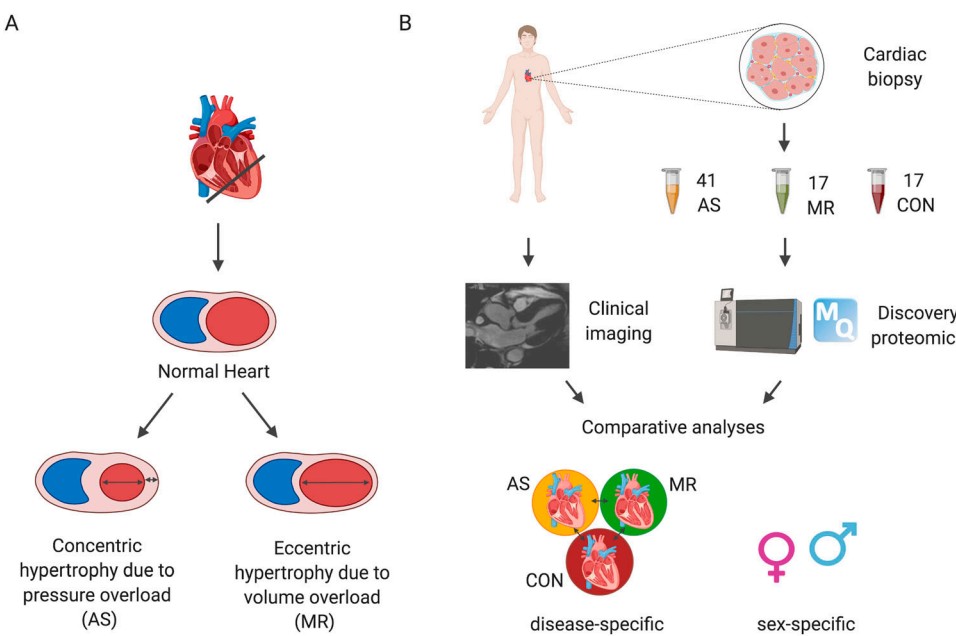

A

B

**Figure 1. Study design.**
**(A)** Aortic stenosis results in pressure overload and leads to concentric cardiac hypertrophy, whereas mitral valve regurgitation causes volume overload and leads to eccentric hypertrophy. **(B)** Proteins were extracted from biopsies of the left ventricle of 41 patients with aortic stenosis (female n = 21; male n = 20), 17 patients with mitral valve regurgitation (female n = 5; male n = 12), and 17 healthy control hearts (CON) (female n = 8; male n = 9). Protein extracts were digested to peptides using trypsin and LysC and analyzed by high-resolution tandem mass spectrometry (LC–MS/MS). A deep reference proteome, acquired by analyzing a high-pH prefractionated peptide mixture of all samples, was used for MS1 matching. RAW data were processed with MaxQuant, and LFQ intensities were used for disease- and sex-specific protein expression analyses. Integrative analyses were performed on proteomic data and clinical imaging (cardiac magnetic resonance imaging) phenotypes.

# Results

## Patient cohort

Our cohort is comprised of 75 human LV myocardial samples (41 AS, 17 MR, and 17 controls). Biopsies were performed during open heart valve replacement surgery in patients with AS and in patients with MR. Study design and patient characteristics are described in Fig 1 and Table 1, respectively. Left ventricular morphology is different between patients with AS and patients with MR seen for LV end-diastolic volume, for example, which is higher in MR (Table 1 and Fig 2). Disease-specific parameters such as degree of MR, mean pressure gradient across the aortic valve, and age at surgery were different between AS and MR (Table 1 and Fig 2). Sex-specific differences in clinical parameters were seen in AS patients by comparing female values against male values but not in MR patients (Fig 3).

## Heart proteome coverage and tissue-specific features

A deep heart proteome dataset was generated from a mixed reference sample consisting of equal parts of AS, MR, and control (CON) samples. Using two-dimensional liquid chromatography before tandem mass spectrometry analysis, we identified, in total, 8,365 distinct protein groups. This deep reference proteome was used, as described by Doll et al (2017), to match MS2 identification across individual runs to peptide precursors and to reach a uniform coverage across all samples with an average of 3,561 (±187) proteins quantified per individual sample (Fig S1A).

Overall, myosin heavy chain 7 (*MYH7*), titin (*TTN*), and actin alpha cardiac muscle 1 (*ACTC1*) represent the first quartile of total cumulative protein intensity. Together with actinin alpha 2 (*ACTN2*), these proteins are in line with the top abundant proteins described

in human heart tissue (Doll et al, 2017). Other proteins typically found in heart tissue such as collagens and heat shock proteins are also quantified robustly (Fig S1B). The principal component (PC) analysis (Fig 4A) already reveals some degree of separation between AS and MR and CON along the first PC, whereas the second PC shows the degree of separation between MR and CON. The AS cohort shows larger differences with controls and higher within-group variability. Although the MR cohort covers patients with moderate and severe MR, no clear separation of these two echo-cardiographically classified disease groups was observed at the proteome level (Fig 4B).

## Quantitative proteome comparison between disease and healthy samples

We describe differentially expressed proteins between disease and healthy states (1,332 [AS versus CON], Fig 5A; 400 [MR versus CON], Fig 5B), and between AS and MR (903 [AS versus MR], Fig 5C) (Fig 5D and Table S1). Notably, more than two-thirds of changes show a decrease in protein expression specifically in AS samples (Fig 5A and C).

Shared effects in protein regulation in AS and MR are seen in 270 proteins, whereas divergent effects (opposing directions of regulation between AS and MR) are seen in five proteins (Fig 5D). 518 changes were found as AS specific (not seen in MR) and 79 as MR specific (Fig 5D). When comparing to the sex-matched control, 468 proteins were significantly differently regulated in female AS, 229 proteins in male AS, 95 proteins in female MR, and 102 proteins in male MR (Fig 5G and H). The full set of differential expression analysis results is summarized in Table S1.

Comparison of our results with a recent proteomic study on AS subtypes with different disease burdens showed excellent agreement in the overlap of quantified proteins in general (Fig S2A) and

**Table 1.  Patient characteristics.**

| Preoperative parameters | AS, n = 41 | MR, n = 17 | *P*-value |
|---|---|---|---|
| Age, yr | 68 ± 9 | 60 ± 14 | 0.03 |
| BMI, kg/m$^2$ | 28 ± 4 | 27 ± 3 | 0.34 |
| Gender balance (female/male) | 21/20 | 5/12 | 0.22 |
| Systolic blood pressure, mm Hg | 140 ± 19 | 131 ± 16 | 0.12 |
| Diastolic blood pressure, mm Hg | 74 ± 11 | 76 ± 14 | 0.67 |
| Hypertension, n (%) | 27 (69) | 11 (65) | 1 |
| Dyslipidemia | 8 (21) | 3 (18) | 1 |
| Diabetes mellitus, n (%) | 7 (17) | 2 (12) | 1 |
| Coronary artery disease, n (%) | 2 (5) | 2 (12) | 1 |
| Atrial fibrillation, paroxysmal | 2 (5) | 2 (12) | 0.709 |
| Atrial fibrillation, permanent | 0 (0) | 2 (12) | 0.149 |
| Left ventricular end-diastolic volume, ml/m$^2$ | 73 ± 17 | 108 ± 35 | <0.001 |
| Left ventricular myocardial mass, g/m$^2$ | 71 ± 20 | 67 ± 15 | 0.385 |
| Mean pressure gradient aortic valve, mm Hg | 56 ± 15 | 4 ± 8 | <0.0001 |
| Mitral valve regurgitation, grade (none/mild, moderate, severe) | (41, 0, 0) | (0, 10, 7) | <0.0001 |
| Aortic valve insufficiency, grade (none/mild, moderate, severe) | (36, 5, 0) | (17, 0, 0) | 0.321 |
| Left ventricular ejection fraction, % | 60 ± 7.4 | 64 ± 6.2 | 0.13 |
| Medication: ACE inhibitor, n (%) | 15 (37) | 5 (29) | 1 |
| Medication: beta-blocker, n (%) | 20 (49) | 10 (59) | 0.358 |
| Medication: diuretics, n (%) | 12 (29) | 5 (29) | 1 |

Data are presented as numbers (%) or mean ± SD. ACE inhibitor, angiotensin-converting enzyme inhibitor; AS, aortic valve stenosis; BMI, body mass index; MR, mitral valve regurgitation; n, number. Mean pressure gradient aortic valve describes severity of aortic valve stenosis, and mitral valve regurgitation describes severity of mitral valve insufficiency. Statistical comparison was performed with a two-sided, two-sample Wilcoxon rank test in case of numeric data and with a chi-squared test in case of categorical data. Controls (n = 17; female = 8, male = 9) were 44 ± 15-yr-old healthy cardiac organ donors without cardiovascular diseases.

also of AS- but not MR-specific significantly regulated proteins (Fig S2B and C) (Brandenburg et al, 2022). The higher numbers of significantly regulated proteins in our study can be explained by the deeper proteomic coverage and also overall larger sample size. GO enrichment analysis resulted in 138 GO terms enriched in AS versus CON, 106 terms enriched in MR versus CON, and 25 terms enriched in AS versus MR. REVIGO summary of GO terms and subsequent manual assignment to five categories reveal changes mainly in the following distinctive categories: (I) ECM composition, (II) energy metabolism, (III) proteostasis, (IV) cytoskeleton, and (V) other terms (Fig 5E and F). We therefore explored disease- and sex-specific effects with regard to these categories in more detail. Enriched GO terms and their category assignments are summarized in Tables S2 and S3. GO enrichment analysis of up- and down-regulated AS- and MR-specific or commonly regulated proteins also confirmed that most of the proteome alterations in our study can be systematically grouped into the four categories described above (Table S4).

## ECM composition

Myocardial ECM composition plays a crucial role in heart diseases (Frangogiannis, 2019). In general, ECM-related proteins were significantly higher in abundance in AS and MR compared with

controls (Fig 6A). When comparing intensity values of all ECM proteins (annotation based on Doll et al [2017]) among the three study groups, we observed higher abundance in AS and MR (Fig 6B). Fibrous tissue content quantified by cardiac magnetic resonance (CMR) was also higher in AS and MR compared with reference values from literature data (Doltra et al, 2014) (Fig 6C). Enrichment analyses show not only a common but also a disease-specific increase in ECM-related GO terms for AS and MR (Fig 6D and Table S2). In AS and MR, non-fibrillar collagens (collagen types 6, 12, 14, and 18) and matricellular proteins (*POSTN* and *TGFBI*) were more abundant, as were ECM glycoproteins (*TNXB* and *FBN1*), proteoglycans (*VCAN* and *HAPLN1*), and three members of the SLRP (small leucine-rich proteoglycan) class, namely, biglycan (*BGN*), decorin (*DCN*), and asporin (*ASPN*). *BGN* and *DCN* have been suggested to regulate the amount of collagen and fibrillogenesis in the heart (Melchior-Becker et al, 2011) (Figs 6A and S3).

AS samples show specific enrichment of distinct collagen-related GO terms (Fig 6D). There is a pronounced higher amount of fibrillar collagens such as collagen type I (*COL1A1* and *COL1A2*), which forms thicker and stiffer fibers, and collagen type III (*COL3A1*), which forms more compliant and elastic fibers (Fig 6A). The level of increase and the changes in the ratio between collagen types I and III are well described for HF in general and patients with dilated

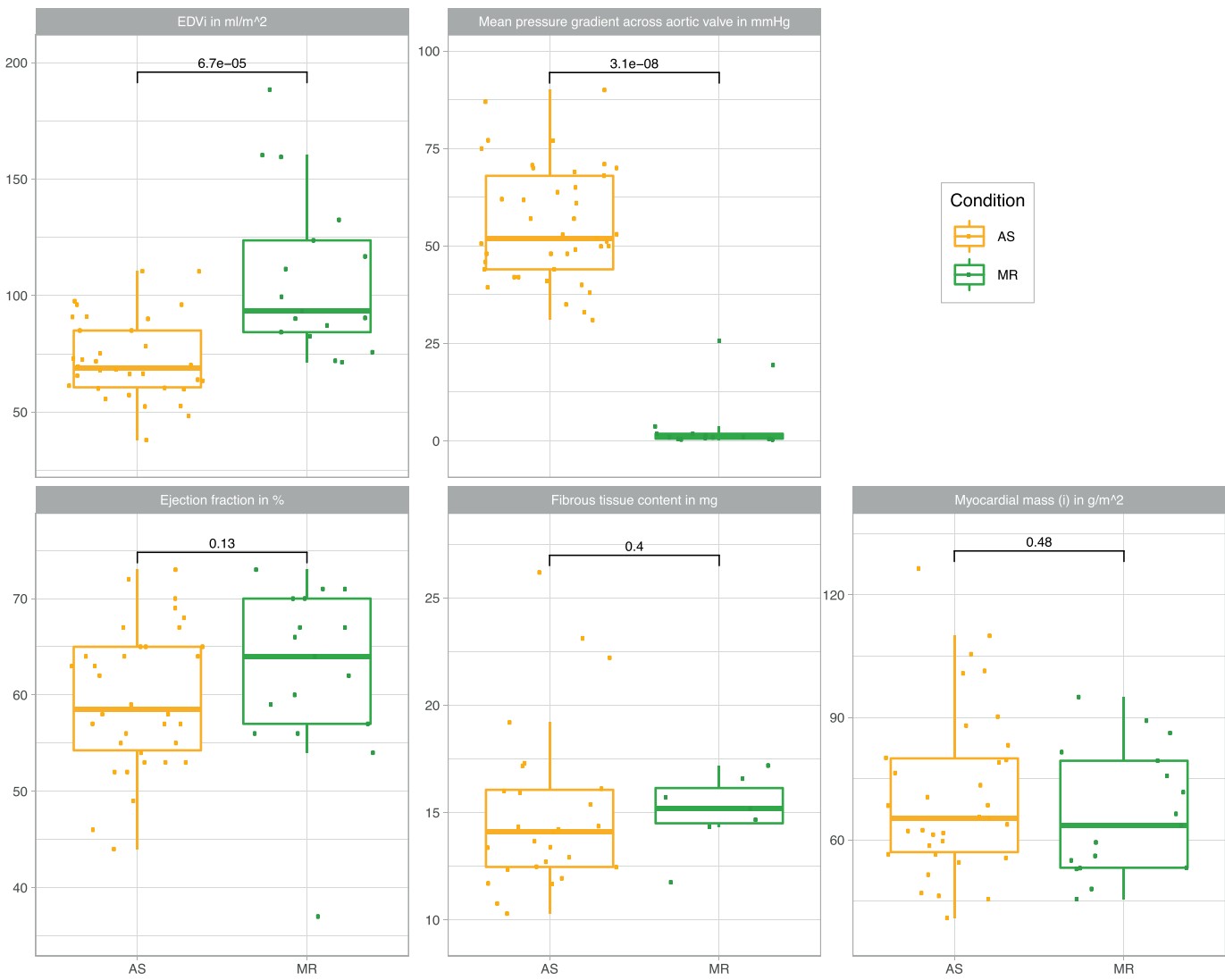

**Figure 2. Imaging parameters.**
Overview of left ventricular clinical parameters from magnetic resonance imaging stratified by conditions (AS/MR). AS, aortic valve stenosis; EDVi, end-diastolic volume indexed to body surface area; MR, mitral valve regurgitation. Statistical comparison was performed with a two-sided, two-sample Wilcoxon rank test.

cardiomyopathy and AS in particular. Here, we detected an additional increase in collagen type V and thrombospondin-4 (THBS4) in AS, which has only been described in animal studies so far (Honda et al, 1993; Mustonen et al, 2008). Thrombospondin-5 (*COMP*) is also specifically up-regulated in AS and may belong to the expression signature of matrifibrocytes, which are derived from *POSTN*-expressing cells and have been shown to form stiff scar tissue in infarcted mouse hearts (Fu et al, 2018). Similarly, *CILP* (cartilage intermediate layer protein), a mediator of cardiac ECM remodeling and a marker for cardiac fibrosis, is up-regulated specifically in AS (Park et al, 2020).

In MR, we see specific up-regulation of enzymatic proteins (*CPA3* [carboxypeptidase A3]; *CMA1* [chymase 1]) and proteins expressed in developing arteries and epithelial cells (*FBLN5* [fibulin 5]; *COL6A6* [collagen type VI alpha 6 chain]). Furthermore, there is an increase in annexin 6 (*ANXA6*), which was shown to regulate the transition

from chronic hypertrophied cardiomyocytes to apoptosis in cell culture (Banerjee et al, 2015).

Enrichment analyses of proteins only increased in male AS and MR revealed terms related to ECM composition (Figs 6E and S4). Among these proteins, we found members of the *ITIH* family, namely, *ITIH1*, *ITIH2*, *ITIH4* (AS), and *ITIH4* (MR), which function as ECM stabilizers (Fig 6A), and fibronectin (FN1) (AS) (Fig 6A), which plays a prominent role in fibrosis and cardiac function according to an HF animal model (Valiente-Alandi et al, 2018). In male AS only, *serpinE2* is elevated, which is in accordance with a previous study, where pressure overload hypertrophy in mice led to up-regulation of *serpinE2* and accumulation of collagens, thus contributing to cardiac fibrosis (Li et al, 2016b). In female AS, we found lower levels of *STAT3*, which is a transcription factor and an important contributor to collagen synthesis and cardiac fibrosis (Mir et al, 2012). Mirroring proteome measurements, we see a higher degree of

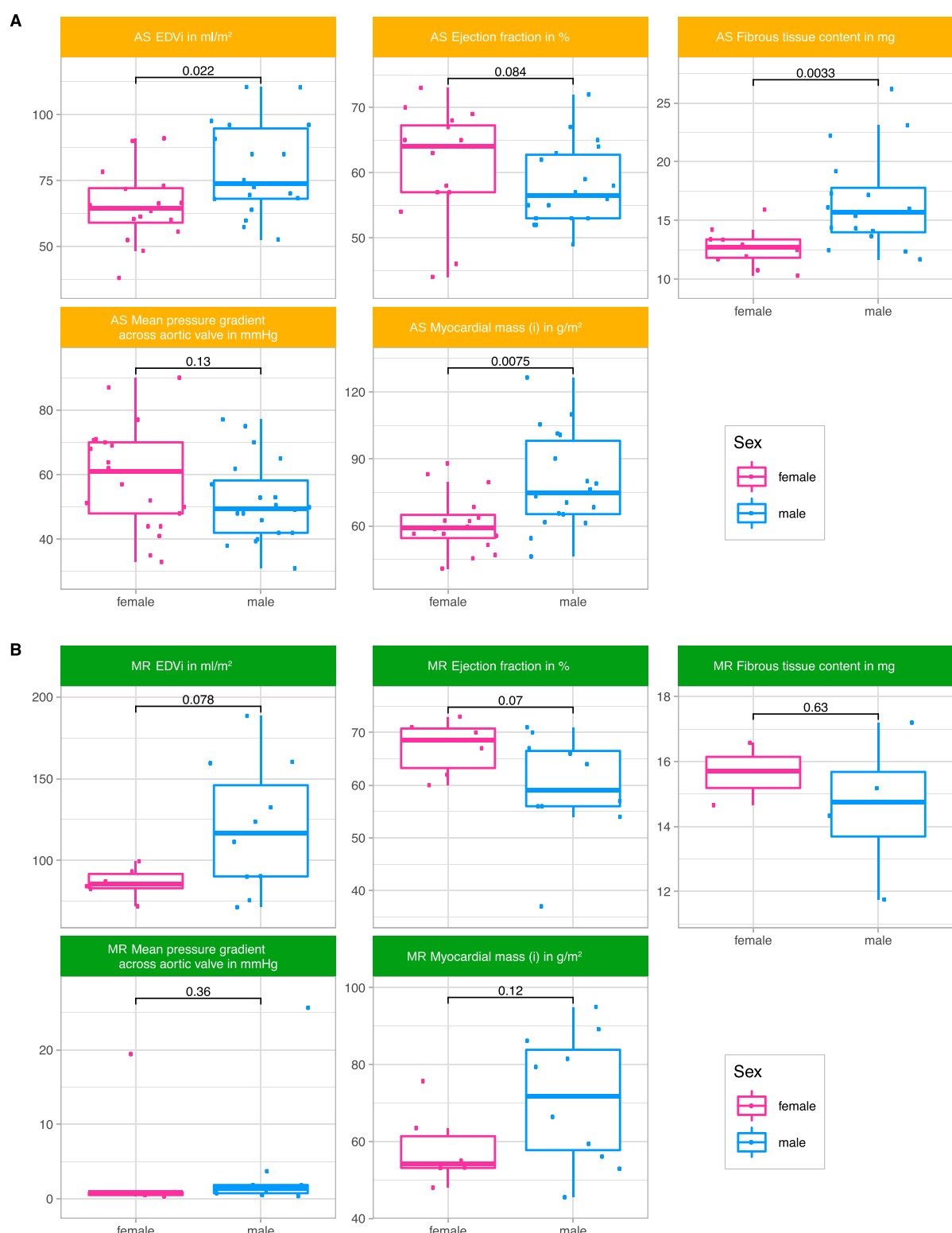

**Figure 3. Sex-stratified imaging parameters.**
Overview of clinical parameters from magnetic resonance imaging stratified by sex in AS and MR patients. End-diastolic volume, ejection fraction, fibrous tissue content, and myocardial mass of the left ventricle are described, and the mean pressure gradient across the aortic valve. AS, aortic valve stenosis; EDVi, end-diastolic volume indexed to body surface area; MR, mitral valve regurgitation. Group-wise statistical comparison was performed with a two-sided, two-sample Wilcoxon rank test.

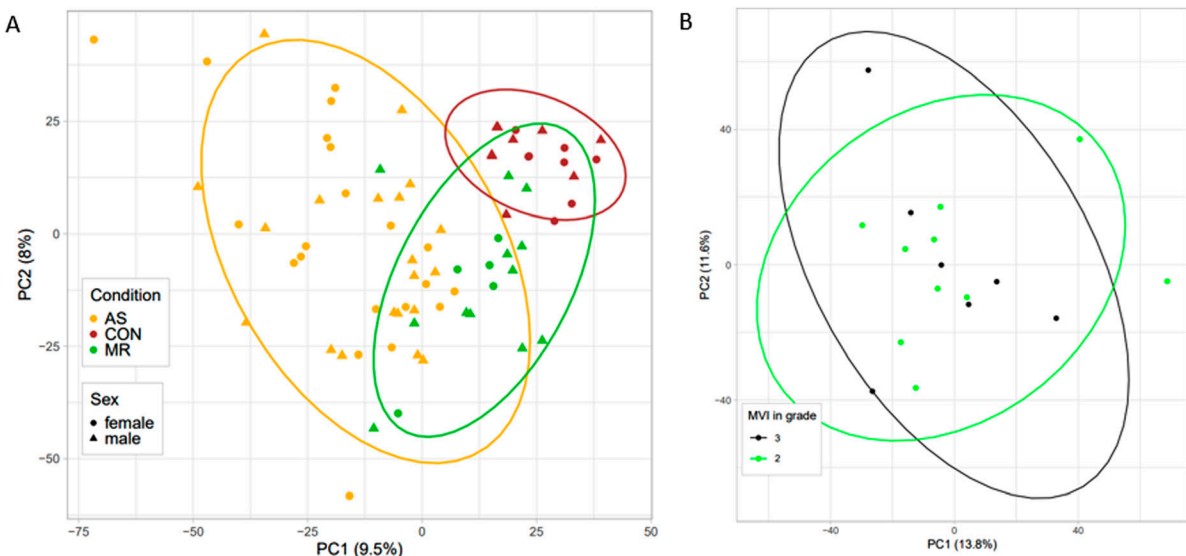

**Figure 4. Heart proteome distribution.**
**(A, B)** Principal component analysis of protein measurements displaying (A) all three conditions and sex assignment and (B) moderate (2) and severe (3) mitral valve insufficiency grades within the patient cohort of MR. AS, aortic valve stenosis; CON, healthy control hearts; MR, mitral valve regurgitation.

fibrous tissue content in male AS compared with female AS in clinical imaging (Fig 6F). All proteins described are shown in Fig S3.

### Energy metabolism and mitochondria

The normal cardiac function relies on a constant high energy supply, which in the healthy heart is mainly provided by oxidative phosphorylation from fatty acid oxidation (Stanley & Chandler, 2002).

Proteins of important metabolic pathways are down-regulated in AS and MR (Fig 7A–C). This includes proteins involved in the tri-carboxylic acid cycle, which were found to be down-regulated in AS and MR (Fig 7B and C). Others, such as the major transporter for long-chain fatty acids (SLC27A6 or FATP6) and the main glucose transporter in cardiac tissue GLUT1 (insulin-independent HepG2 glucose transporter, SLC2A1), are up-regulated in AS and MR (Figs 7C and S5). Changes in metabolic proteins were more pronounced in AS; especially, proteins involved in fatty acid β-oxidation and branched-chain amino acid catabolism showed AS-specific reduction (Fig S6). Also, the median abundance of proteins assigned to mitochondria (Doll et al, 2017) was significantly reduced in AS but not in MR (Fig 7D). Changes in myocardial energy supply can affect cardiac function, which can reliably be measured as ejection fraction in magnetic resonance imaging (Woodard et al, 2006). In our cohort, AS patients showed a tendency toward stronger reduction in ejection fraction compared with normal values than MR patients (Fig 7E).

Sex-stratified analysis showed differences between AS and MR patients. In AS compared with MR, male-specific down-regulation of metabolic proteins was more pronounced, mainly considering mitochondrial matrix proteins (Fig 7F). Regarding the clinical parameter ejection fraction, female and male AS patients showed significant reduction compared with their respective normal values,

whereas in MR, only male patients showed significant reduction. In line with the more pronounced sex differences in down-regulated proteins involved in energy metabolism in AS compared with MR, female AS patients showed more differences in clinical parameters of cardiac remodeling (end-diastolic volumes, myocardial mass, and fibrous tissue content) compared with male AS patients than female MR patients compared with male MR patients (Fig 3).

Alterations more pronounced in AS include higher abundance of PYGB, a protein responsible for glycogen degradation, and abundance changes for two glycolysis-related proteins (PFKM and ENO2) (Fig 7A). The metabolic regulator PDK4 is decreased 13-fold in AS, but only sixfold in MR (Fig 7C). PDK4 phosphorylates and thus inactivates pyruvate dehydrogenase. Less inactivation of pyruvate dehydrogenase can contribute to increased glucose oxidation and as a result pyruvate use for acetyl-CoA generation. Furthermore, sirtuin-3 (SIRT3) is significantly reduced in AS compared with control and MR (Fig 7A) and down-regulation of sirtuins has been described in mitochondrial dysfunction found in pathological hypertrophy (Matsushima & Sadoshima, 2015).

In contrast, TSPO (translocator protein) is down-regulated in MR only (Fig 7A). TSPO belongs to the mitochondrial cholesterol/porphyrin uptake translocator protein family, found to be up-regulated in pressure-overloaded hearts in mice. Preventing TSPO increase limits the progression of HF, preserves ATP production, and decreases oxidative stress, thereby preventing metabolic failure (Thai et al, 2018). Although not statistically significant (P = 0.07), TSPO expression is increased twofold in AS, which is consistent with observations in pressure overload by Thai et al (2018). However, down-regulation in MR suggests a divergent mechanism in volume overload.

An example of common up-regulation in AS and MR is SPTLC (Fig 7C), an enzyme crucial in the de novo synthesis of ceramides.

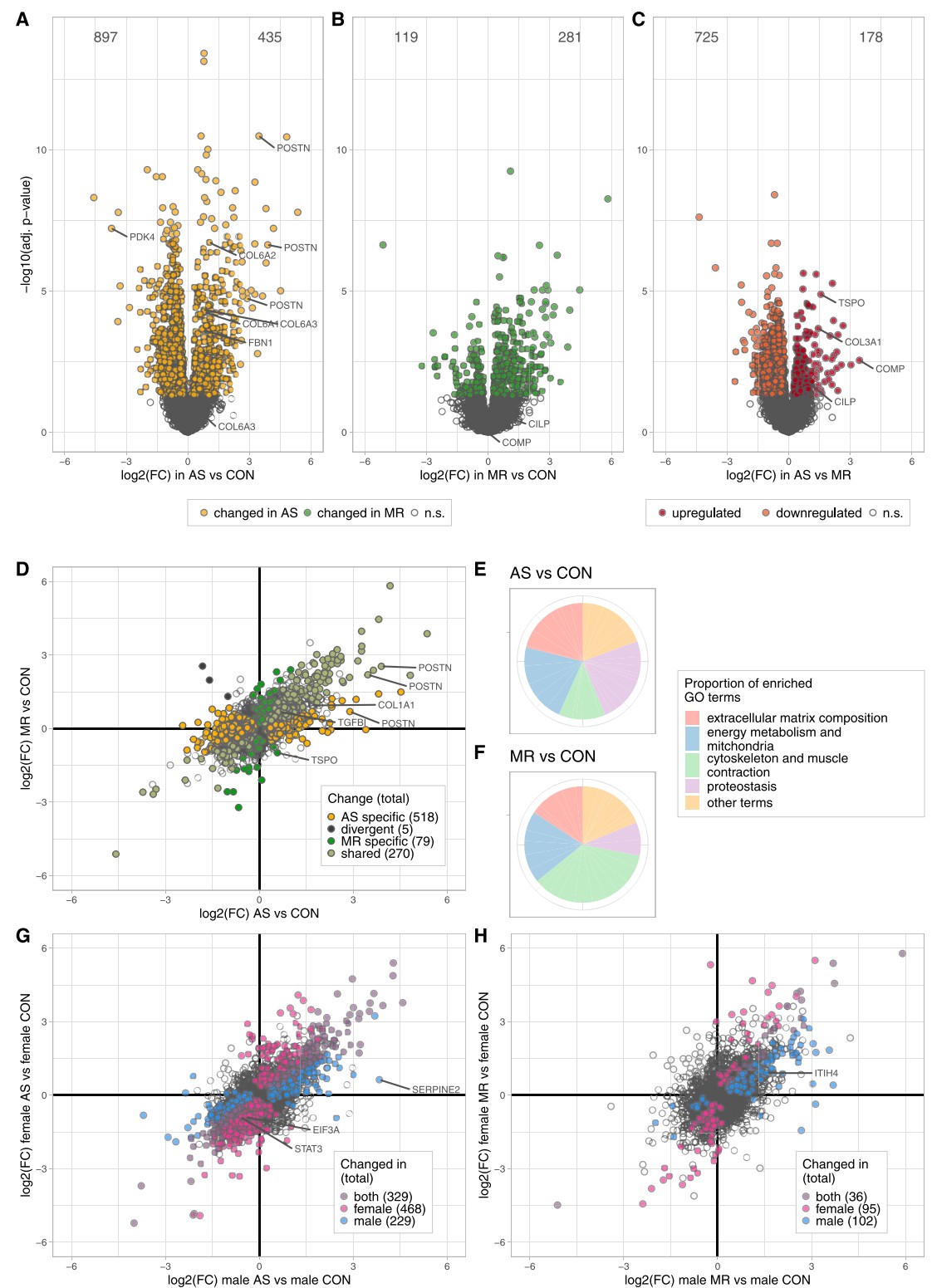

**Figure 5. Quantitative analyses of disease- and sex-specific differences in protein abundance.**
**(A, B, C)** Volcano plots denoting the log$_2$ fold change and the negative log$_{10}$ of the Benjamini–Hochberg-adjusted *P*-values for the comparisons between AS versus CON (A), MR versus CON (B), and AS versus MR (C). Differentially expressed proteins (adjusted *P*-value < 0.05) are colored in yellow (AS) or green (MR) and red (up-regulated in AS compared with MR) and orange (down-regulated in AS compared with MR). **(D)** Scatter plot of log$_2$-fold changes in AS and MR versus CON. Colors denote changes that are shared (pale green) or divergent in the comparison against CON (black) or the specific one (yellow for AS; green for MR); that is, the change against CON is also significant against the respective other condition. Statistical analysis was performed using R including limma linear models and the Benjamini–Hochberg multiple testing correction. **(E, F)** Pie chart of GO term enrichment analysis. The merged set of enriched terms from up- and down-regulation is summarized into redundant clusters by

Overexpression leads to accumulation of ceramides and subsequently to changes in the lipid profile, apoptosis, reduction in oxidative metabolism, and progression of maladaptive remodeling (Ji et al, 2017).

### Proteostasis

Proteostasis describes a balance of biological pathways including protein synthesis, folding, quality control, trafficking, and clearance, which ensures proper cell function (Henning & Brundel, 2017). In AS, proteins related to proteostasis were significantly reduced, which was only seen partly in MR (Fig 8A).

We found down-regulation of proteins belonging to GO terms describing translation including ribosomes, protein folding and quality control, that is, chaperonin containing T-complex protein Ring Complex (TRiC), and trafficking, such as protein targeting to endoplasmic reticulum in AS (Figs 8B and S7). Notably, all subunits of the TRiC (TCP-1, CCT2–7) are lower in abundance in AS when compared with controls and to MR (Fig 8A). The complex is required for proper folding of essential cytoskeletal proteins, such as actin and tubulin. One major Nedd8-conjugating enzyme, UBE2M, is up-regulated in AS (Figs 8A and S7), known to modify cullin scaffold proteins and to activate the SCF ubiquitin ligase complex (Skip, Cullin, F-Box), which is crucial for degradation of many target proteins via the ubiquitin–proteasome system. Cullins (CUL-1, CUL-4, and CUL-5), however, are lower in abundance in AS patients (Fig 8A). Similarly, several subunits of the COP9 signalosome subunits (COPS3–5), which have a deneddylase activity and remove Nedd8 from the Cullin scaffold, are also down-regulated in AS (Fig 8A). The disturbance of protein homeostasis may lead to well-known cardiac proteotoxicity.

Down-regulation of proteostasis is more pronounced in female AS. GO term enrichment shows a strong signal for translation and protein targeting (Fig 8C). 13 of 31 detected 40S and 27 of 42 60S ribosomal subunits are changed in females only; another nine ribosomal subunits are down-regulated in both.

In addition, we found 16 out of 39 detected subunits spanning all cytosolic eukaryotic initiation factor complexes to be less abundant in females compared with none regulated in males. However, in females, mainly cytosolic enzymes are reduced, whereas in males, mitochondrial ribosomal units are less abundant (Fig S8). Knockdown of EIF3a has been shown to ameliorate cardiac fibrosis and shows in our study a 1.9-fold female-specific down-regulation (Li et al, 2016a).

The reduced abundance of ribosomal and translation-related proteins suggests a reduced cytosolic protein synthesis capacity in female AS. Protein synthesis capacity is required for cardiac hypertrophy progression and has been discussed as a therapeutic target to prevent or reduce cardiac hypertrophy (Blackwood et al, 2020). Female AS patients in this cohort show less increase in LV mass (hypertrophy) than male AS patients, when compared with the corresponding healthy reference values (Fig 8D), which might be because of reduced protein synthesis capacity.

Heat shock proteins have been described to play a role in cardiac hypertrophy (Kumarapeli et al, 2008). The most abundant small heat shock protein in cardiomyocytes αB-crystallin (CRYAB), which has been shown to suppress pressure overload cardiac hypertrophy in mice (Kumarapeli et al, 2008), was found down-regulated in AS and MR (Fig 8A). Heat shock protein β-7 (HSPB7) is a cardioprotective stabilizer for large sarcomere proteins, and its loss leads to autophagic compensation by degrading accumulated protein aggregates (Mercer et al, 2018). We detected significantly less HSPB7 (isoforms 1 and 2) abundance in AS and MR (Fig 8A and Table S5). Furthermore, Hsp70 (HSPA1B), for which a gene knockout has been described to induce cardiac dysfunction and development of cardiac hypertrophy, is down-regulated in both disease conditions (Fig 8A) (Kim et al, 2006). TRAP1, known to protect the heart from hypertrophy, was found down-regulated in AS samples only (Fig 8A) (Zhang et al, 2011).

### Cytoskeletal, adhesion, and contractile proteins

The cytoskeleton plays a crucial role in maintaining cellular stability and reacting to mechanical stresses through signal transmission and subsequent remodeling. In general, we observe a strong increase in cytoskeletal and contractile proteins with disease-specific variations (Figs 9 and S9). Sex differences are less evident (Fig 9A).

Proteins belonging to actin binding and to myofibrils are increased in AS and MR (Fig 9B) with shared up-regulation of proteins involved in anchoring actin filaments to the membrane (filamin A [FLNA] and alpha-actinin-1), and intermediate filaments such as vimentin (VIM), synemin (SYNM), and the nuclear lamin A/C (LMNA), acting as bridges between cell organelles and sarcolemma (Fig 9A). In addition, many contraction-associated proteins (tropomyosin [TPM1, TPM3] and troponins [TNNI1, TNNT2]) are higher in abundance, whereas others (MYH7, MYL5, MYL12B, and SMPX) are down-regulated (Fig 9A). Interestingly, many changes in actomyosin proteins may be assigned to non-sarcomeric structures such as smooth muscle cells (MYL12B, MYH11, and CNN3) or to non-muscle myosin 2B (MYH10), for which an impact on cardiac remodeling has been described when increasingly deposited at costameres in animals (Pandey et al, 2018).

Specifically in AS, alpha-integrins (ITGA5, ITGA6, and ITGAV) are down-regulated, as was melusin (ITGB1BP2), which was found to have a protective effect in chronic pressure overload (Tarone & Brancaccio, 2015). Two intermediate filament proteins are increased in AS, namely, nestin, described to be expressed during early heart development only (Sejersen & Lendahl, 1993), and desmin, the major connector of costameres, desmosomes, myofibrils, and nucleus, and found to be up-regulated in HF (Chen, 2018). A multitude of proteins involved in actin bundles and stress fiber formation (FSCN1, PDLIM4, and MARCKS) and in transducing mechanical stress signals toward the nucleus (TRIP6, ZYX, ABLIM1, and

---

REVIGO. **(E, F)** Proportions are based on the GOA frequency of the cluster representative terms, which are assigned to five categories for AS versus CON (E) and MR versus CON (F). **(G, H)** Scatter plots of log$_2$ fold changes in AS and MR stratified by sexes. Colors denote changes only significant in either female, male, or both versus their sex-matched controls. AS, aortic valve stenosis; CON, healthy control hearts; MR, mitral valve regurgitation; ns, not significant.

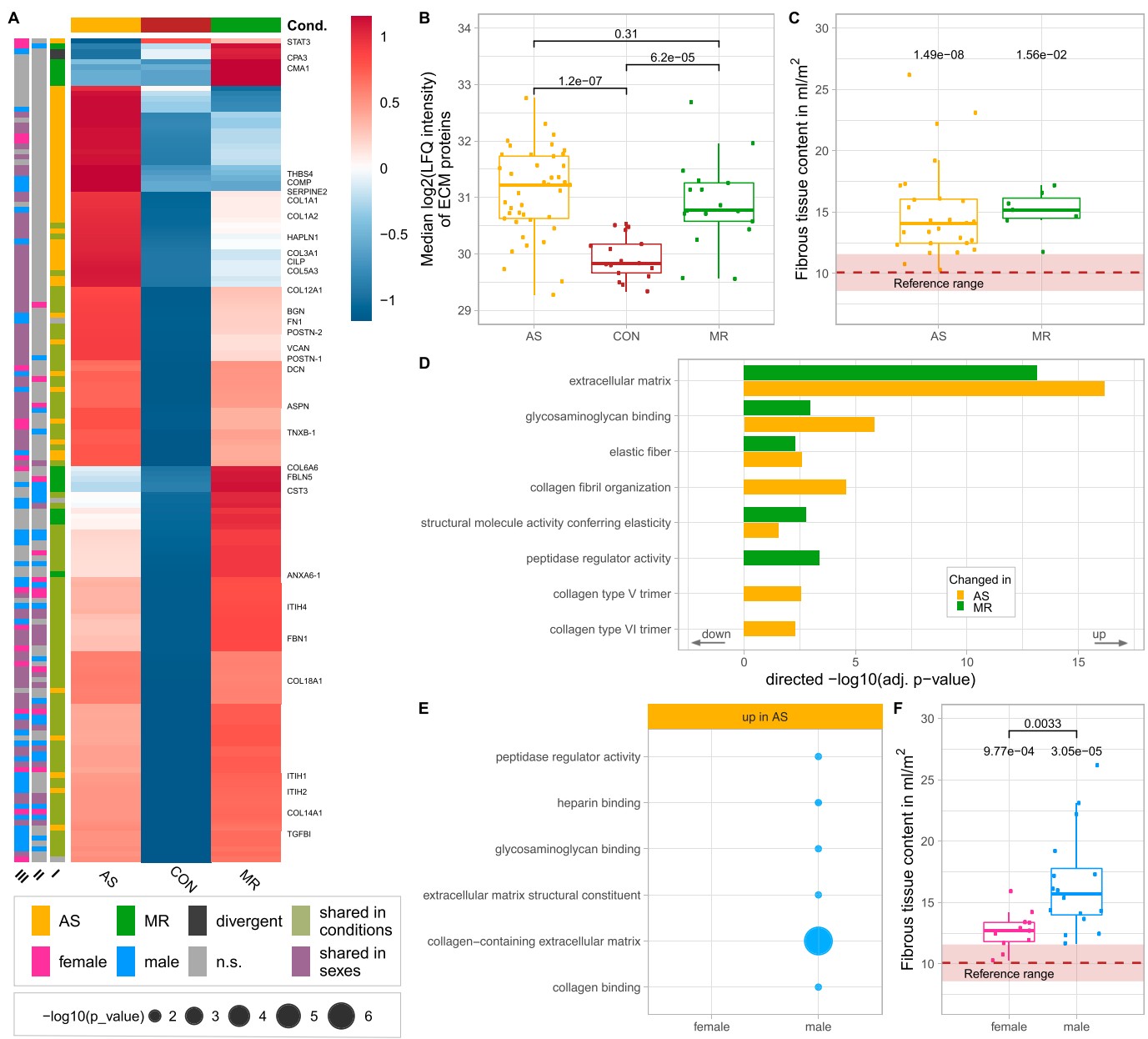

**Figure 6. Disease- and sex-specific differences in abundance of proteins related to ECM composition.**
**(A)** Clustered heatmap showing the condition's mean abundance of proteins belonging to ECM-related GO terms. Annotation bars denote significant changes in condition (I) and in sex (II—effect in sex MR; III—effect in sex AS). Proteins described in the text are labeled. **(B)** Median log$_2$ of LFQ intensity of all proteins belonging to the ECM; $P$-values are calculated using a Wilcoxon rank test. **(C)** Comparison of fibrous tissue content as measured by MRI in AS and MR (C). $P$-values are calculated using a Wilcoxon rank test with one sample against the reference mean. Dots represent individual subjects. **(D, E)** Combined results (−log$_{10}$-transformed $P$-value) from GO term enrichment analysis in AS and MR versus CON (D) shown in the direction of regulation they were found in and on proteins found increased in male or female AS only (E). **(F)** Comparison of fibrous tissue content as measured by MRI in AS stratified by sex. $P$-values are calculated using a Wilcoxon rank test with two samples (female versus male, denoted by brackets). AS, aortic valve stenosis; CON, healthy control hearts; MR, mitral valve regurgitation; ns, not significant.

*SEPT7*) are also increased in AS (Fig 9A). The major myosin heavy chain isoforms *MYH6* and *MYH7* and heavy and light chains previously not believed to be expressed in cardiac tissue (*MYL5*, *MYH4*, and *MYH8*) are less abundant in AS. Altered levels of *MYH6* and *MYH7* manifest in a significantly lower ratio between the two sarcomeric heavy chains (*MYH6*/*MYH7*) only in AS (Fig 9C), which is compliant with literature (Reiser et al, 2001). Specific changes in AS are summarized in Fig S10.

MR-specific changes cover cell–matrix adhesion (costamere) and cytoskeletal proteins just beneath the sarcolemma (Fig 9D and E). Members of the dystrophin–glycoprotein complex, dystrophin (*DMD*), a dystrophin-binding protein (*SNTB2*), and a membrane-spanning protein (*SGCE*), are higher in abundance in MR. Actin-binding proteins up-regulated only in MR are mainly anchoring proteins such as *SPTBN1*, *SPTB*, *TLN2*, and *ADD3* (Fig 9A and E). Furthermore, up-regulation of actin-binding non-muscle *α*-actinin

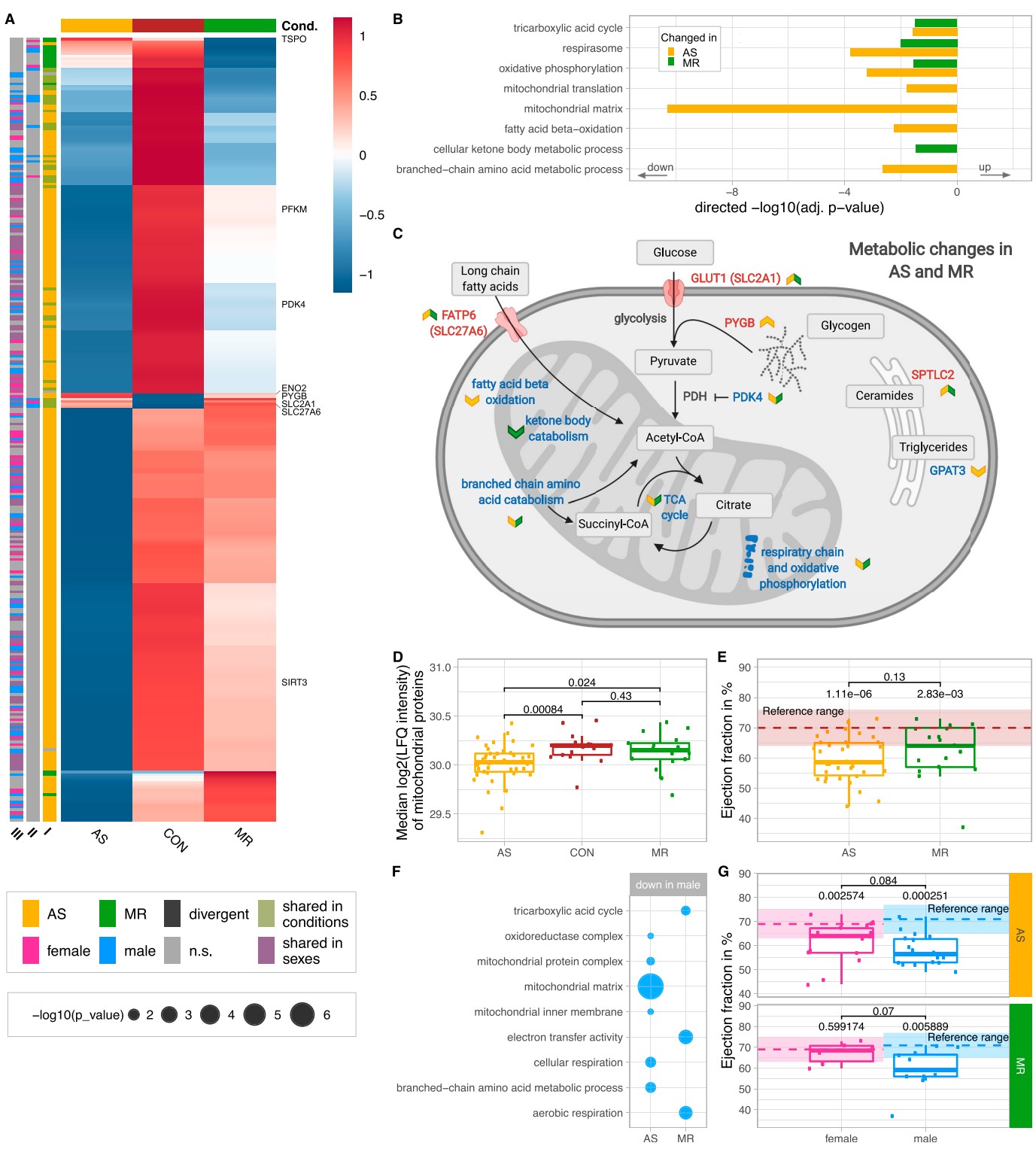

**Figure 7. Disease- and sex-specific differences in abundance of proteins related to energy metabolism and mitochondria.**
**(A)** Clustered heatmap showing the condition's mean abundance of proteins belonging to energy metabolism and mitochondria-related GO terms. Annotation bars denote significant changes in condition (I) and in sex (II—effect in sex MR; III—effect in sex AS). Proteins described in the text are labeled. **(B)** Combined results (−log10-transformed P-value) of GO term enrichment analysis in AS and MR versus CON reduced to representative terms. Bars are shown in the direction of regulation they were found in. **(C)** Summary of metabolic processes derived from enrichment results and selected key proteins. Colored arrows denote direction of change and the condition the change was found in. **(D)** Median log2 of LFQ intensity of all proteins with mitochondrial annotation. P-values are calculated using a Wilcoxon rank test. **(E)** Comparison of ejection fraction (in %) as measured by magnetic resonance imaging in AS and MR. P-values are calculated using a Wilcoxon rank test with two samples (AS versus MR, denoted by brackets) and one sample against the reference mean (no bracket). The reference range consists of the mean (dotted line) ± one SD. For (E), we show

ACTN4 provides evidence of reactivation of fetal actinin forms as was described in failing hearts (Cetinkaya et al, 2020). Among proteins involved in muscle contraction, *MYH2* and *MYH14* showed higher levels, whereas *MYH7B* showed lower levels in MR. Up-regulation of *MYLK3*, a cardiac-specific myosin light chain kinase, may have a positive effect on contractility in MR (Chan et al, 2008). Distinct changes are also evident in proteins involved in calcium handling (Fig S11) and at the desmosome (Fig S12). GO term enrichment analyses of sex-specific enriched proteins resulted in a single hit of the term "actin binding" for male MR samples.

## Discussion

In this study, we present proteome profiles from LV myocardium of female and male patients with severe AS or MR. Together with corresponding clinical and imaging data, they provide detailed insight into human cardiac remodeling processes at both the organ and cellular levels. In our comparative analysis, we found disease-specific and sex-specific differential expression levels of myocardial proteins, mainly in four functional categories: ECM, energy metabolism, cytoskeleton, and protein synthesis. Our work thus provides a comprehensive overview of proteome regulation in chronic pressure and volume overload that also confirms findings from many individual animal models in one single human study. Therefore, the work constitutes a valuable basis for future analyses of cardiac function in health and disease in human studies and in hypothesis-driven preclinical research including search for potential druggable treatment targets. Valvular heart disease is one of the most important causes of HF (Bouma et al, 1999; Nkomo et al, 2006). In the aging population, their importance is likely to increase further. Therefore, it is crucial to have a clear understanding of the cardiac remodeling processes associated with chronic pressure and/or volume overload. This must include the significant sex differences in LV hypertrophy and prevalence of HF because of valvular heart disease that are also not fully understood, yet (Petrov et al, 2010). With this knowledge, it would be possible to better assess the course of the disease and thus optimize the timing of valve interventions and/or pharmacological treatment concepts. However, it was not the scope of the current study to investigate drug effects on the proteome, even though this is a potentially interesting question.

AS and MR patients in this study are good representatives of their disease. AS patients suffer from LV pressure overload because of the increased gradient across the diseased aortic valve, and patients with MR suffer from volume overload because of MR, which leads to an increase in LV end-diastolic volumes. These parameters are significantly different between AS and MR patients, whereas other potentially relevant clinical parameters, such as incidence of diabetes or type of medication, were not different between groups.

We performed an additional analysis studying group size–dependent influence on a number of differentially expressed proteins of the AS group, which showed most alterations in the AS group independent of group size (Fig S13). We can thus conclude that the higher number of altered proteins in the AS group is not an exclusive effect of higher power because there is more regulation when equal sample sizes compared with MR are considered. In addition, we want to mention that in the MR group, there are patients with echocardiographically classified moderate MR and those with severe MR, which might have been a reason for more intra-group variability; however, the principal component analysis showed very similar protein abundance variation (Fig 4B). The AS group, however, shows more intra-group variability with only patients with severe AS. This we believe could result from more pronounced sex differences in pressure overload hypertrophy described in the literature than in volume overload hypertrophy and other factors such as serum dihydrotestosterone levels, which are associated with different degrees of LV remodeling in pressure overload conditions (Petrov et al, 2014; Schafstedde et al, 2022).

Furthermore, there is an age difference between the three groups, which might have an influence on the results. The difference in age between AS and MR and especially between controls and patients is a limitation of the study; however, because of the very limited availability of LV myocardial samples, a healthy control group matching in age was not possible to achieve.

The age of AS individuals spans from a minimum of 41 yr to a maximum of 81 yr and thus covers a fairly large range. Differential abundance analysis using a linear modeling strategy of abundance in relation to age within the condition did not find any significant up- or down-regulation of protein abundance. The same was true for a comparison of protein abundance with age in MR only (29–79 age span) (data not shown). In addition, in AS and MR, the age is homogeneous between sexes. As such, a sole impact of age in our comparison of conditions is unlikely as, for example, the proteostasis effects are strongest in female AS. From human autopsies, an increase in collagen content, for example, is reported between 20–25 yr and 67–87 yr of age, and in 80-yr-old subjects, an increase in collagen I and a decrease in collagen III were found (Meschiari et al, 2017). In our study, we also see an increase in ECM proteins; however, in AS patients, we found a specific increase in collagens I and III, for example, which we did not find in MR patients, suggesting a disease-specific rather than an age-specific expression pattern.

Pressure and volume overload leads to an increase in muscle mass and ECM, which is associated with adverse clinical outcome (Puls et al, 2020). In line with previous studies, we found similar levels of hypertrophy and fibrous tissue content on clinical imaging in AS and MR compared with controls. Proteome profiles, however, reveal differences in protein regulation between AS and MR and between female and male patients. In AS, a more pronounced up-regulation of proteins, which are responsible for an increase in myocardial stiffness, such as collagens I and III, was found.

---

averaged values from female and male ejection fraction. Dots represent individual subjects. **(F)** Combined results (–log$_{10}$-transformed *P*-value) from sex-stratified GO term enrichment analysis. Enriched terms are found in male AS and MR only. **(G)** Comparison of ejection fraction (in %) as measured by magnetic resonance imaging in AS and MR stratified by sex. *P*-values are calculated using a Wilcoxon rank test with two samples (female versus male, denoted by brackets) and one sample against the reference mean (no bracket). The reference range consists of the mean (dotted line) ± one SD. AS, aortic valve stenosis; CON, healthy control hearts; Cond., condition; MR, mitral valve regurgitation; ns, not significant.

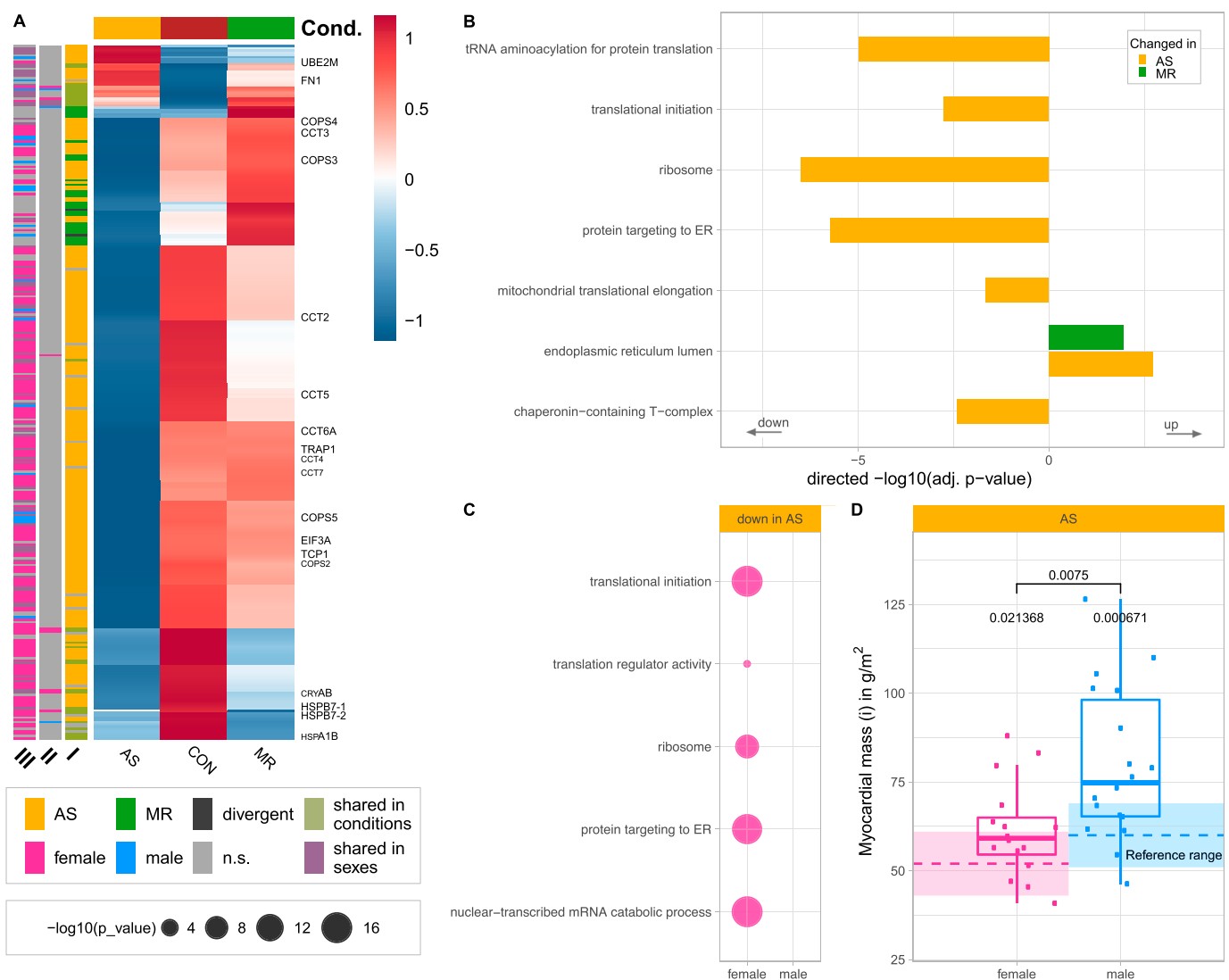

**Figure 8. Disease- and sex-specific differences in abundance of proteins related to proteostasis.**
**(A)** Clustered heatmap showing the condition's mean abundance of proteins belonging to proteostasis-related GO terms. Annotation bars denote significant changes in condition (I) and in sex (II—effect in sex MR; III—effect in sex AS). Proteins described in the text are labeled. **(B, C)** Combined results of GO term enrichment analysis in AS and MR versus CON shown in the direction of regulation they were found in (B) and on proteins found only in one sex of a condition (C). **(D)** Comparison of indexed myocardial mass as measured by MRI in AS stratified by sex. *P*-values are calculated using a Wilcoxon rank test with two samples (female versus male, denoted by brackets) and one sample against the reference mean (no bracket). AS, aortic valve stenosis; CON, healthy control hearts; Cond., condition; MR, mitral valve regurgitation; n.s., not significant.

MR-specific up-regulation was rather seen for enzymatic proteins and proteins related to vessel formation. It is known from human studies that chronic pressure overload triggers profibrotic activation, an increase in ECM, and myocardial stiffness, and fibrillar collagens give structural support to the matrix important for mechanical strength and are described to be up-regulated in myocardial fibrosis (Fan et al, 2012; Frangogiannis, 2019). Unlike pressure overload, the effect of volume overload on ECM remodeling is mainly described in animal models so far and includes activation of proteases, degradation of interstitial ECM, increase in vessel formation, and ventricular dilation (Ryan et al, 2007; You et al, 2018; Frangogiannis, 2019).

Efficient energy metabolism is essential for assuring proper cardiac function, especially in cardiac hypertrophy and HF. Changes

in metabolic remodeling have been described to be associated with the degree of hypertrophy (Ritterhoff et al, 2020). In addition, HF ought to be associated with a reduction in oxidative phosphorylation and an increase in glycolysis for ATP generation (Doenst et al, 2013; Sankaralingam & Lopaschuk, 2015). In our cohort, both AS and MR show reduced fatty acid β-oxidation and an increase in the glucose transporter *GLUT1*. However, specifically in AS, a more pronounced down-regulation of proteins involved in energy metabolism is seen, for example, down-regulation of sirtuin-3, a protein associated with mitochondrial dysfunction and pathological hypertrophy (Matsushima & Sadoshima, 2015). Furthermore, male AS patients show a stronger decrease in proteins involved in energy metabolism than female AS patients, which is consistent with transcriptome studies in mice, describing down-

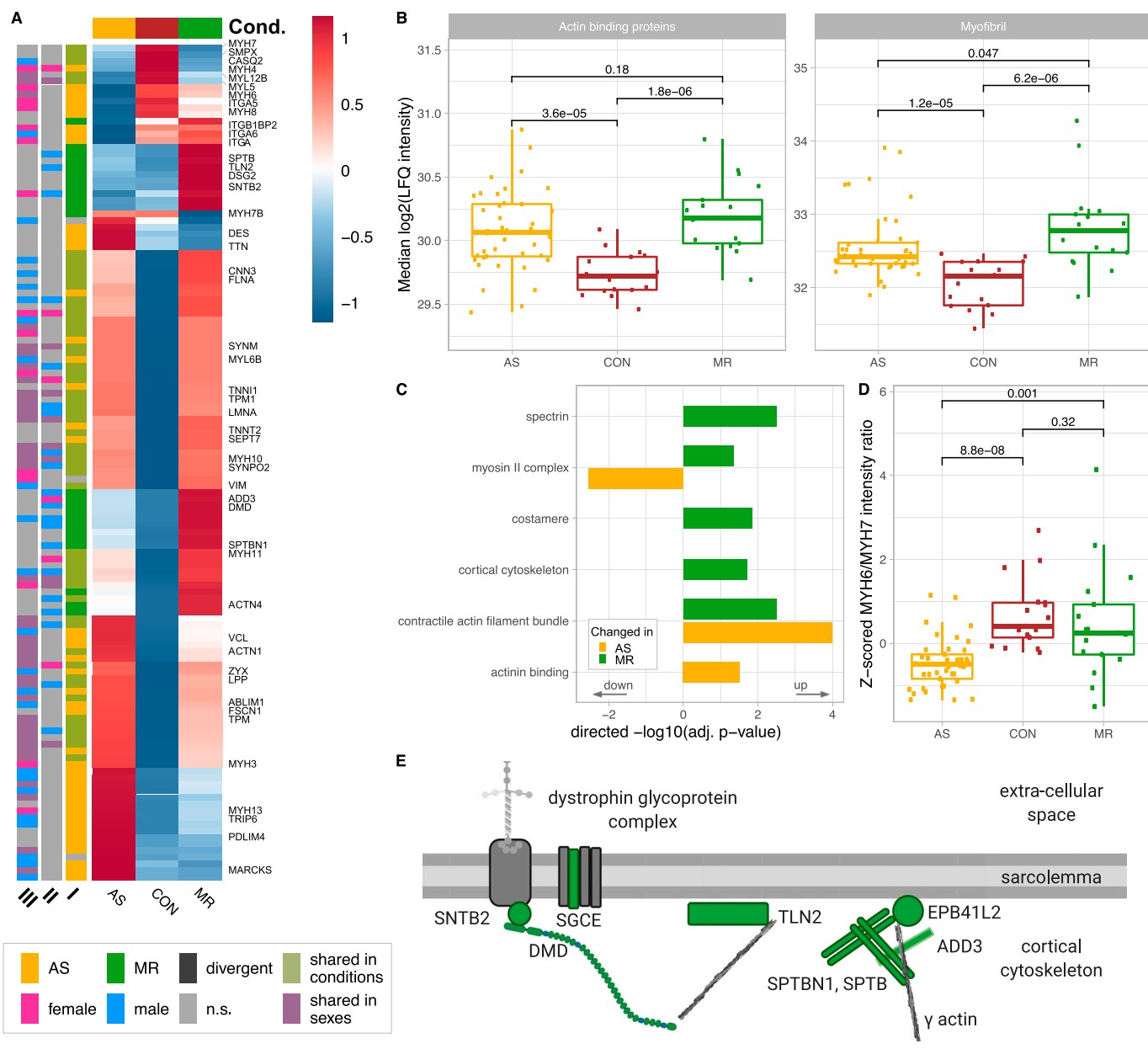

**Figure 9. Disease- and sex-specific differences in abundance of proteins related to cytoskeleton and muscle contraction.**
**(A)** Clustered heatmap showing the condition's mean abundance of proteins belonging to cytoskeleton- and muscle contraction–related GO terms. Annotation bars denote significant changes in condition (I) and in sex (II—effect in sex MR; III—effect in sex AS). Proteins described in the text are labeled. **(B)** Median log$_2$ of LFQ intensity of all proteins with actin binding (left) or myofibril (right) annotation. *P*-values in (B) are calculated using a Wilcoxon rank test. **(C)** Combined results (−log$_{10}$-transformed *P*-value) of GO term enrichment analysis in AS and MR versus CON shown in the direction of regulation they were found in. **(D)** Protein ratio of myosin heavy chains MYH6 and MYH7 in AS, MR, and CON. *P*-values in (D) are calculated using a Wilcoxon rank test. **(E)** Graphical illustration of regulated proteins belonging to the cortical cytoskeleton. The color indicates MR-specific increase (green). AS, aortic valve stenosis; CON, healthy control hearts; Cond., condition; MR, mitral valve regurgitation; ns, not significant.

regulation of the oxidative phosphorylation pathway in male, but not in female, ventricles, and less down-regulation of genes with mitochondrial and respiratory function in females (Kararigas et al, 2014).

Proteostasis describes the maintenance of healthy protein homeostasis. In the heart, this is of particular importance because of the limited regeneration potential of the myocardium (Henning & Brundel, 2017). Proteins involved in protein homeostasis are also associated with the development of cardiac hypertrophy and possibly HF (Hannan et al, 2003; Li et al, 2016a). The proteome profiles of our study showed that proteostasis-associated proteins are down-regulated in AS, affecting many ribosomal subunits and eukaryotic initiation factors, ultimately decreasing protein translation capacity. This down-regulation was almost only seen in female AS patients also having less myocardial hypertrophy compared with men.

Changes in cytoskeletal and muscle contraction proteins are part of pathological remodeling processes and can also cause cardiac dysfunction (Sequeira et al, 2014). In our study, we found an increase in cytoskeletal and contractile proteins in AS and MR. In MR, proteins, which contribute to anchoring the cytoskeletal actin to the sarcolemma and to cross-linking between cytoskeletal entities, are specifically increased, and thus, structural integrity to the cell is provided. Furthermore, alterations in the glycoprotein complex and desmosomal changes point to an increased interconnectedness also toward ECM and neighboring cardiomyocytes, which may be an adaptation to increased stretch caused by volume load. In AS, among the actin-binding proteins, many LIM domain–containing proteins were found, which can display a role in transducing mechanical stress toward the nucleus. In addition, proteins promoting actin bundles and stress fibers are more prominent in AS. This makes sense because pressure load in the heart may have a stronger effect on mechanosensing than volume load. Changes in the *MYH6*/*MYH7* ratio are a common marker for a switch to fetal gene expression as a response to AS and could be confirmed in our cohort (Reiser et al, 2001).

Sex-specific differences in heart valve diseases have been reported to have an impact on remodeling, outcome, and therapy planning (Kararigas et al, 2014; Bienjonetti-Boudreau et al, 2021). Female AS patients have been described to have better preserved cardiac function and less hypertrophy and fibrous tissue content than male AS patients (Petrov et al, 2014). It has been speculated that a less pronounced induction in collagen remodeling contributes to these findings (Kararigas et al, 2014). Our study is the first to apply deep proteomic profiling in female and male human AS patients, and we can confirm a less pronounced upregulation of ECM proteins in female AS. In addition, we observed strong reduction in proteostasis-related proteins and less decrease in proteins involved in energy metabolism in female AS, which could also render a molecular explanation for this clinical observation.

In MR patients, we have detected sex differences at the level of clinical parameters with less hypertrophy, less dilation, and better LV function in female compared with male patients. Proteome profiling revealed that in women, ECM proteins were less exclusively up-regulated and metabolic proteins were less down-regulated. These cellular adaptation processes would thus also fit well with the phenotype and clinical picture of the patients. In general, the gender balance analysis rendered fewer altered proteins than the one with all subjects. The smaller number of subjects in the gender balance analysis and differences in regulation between females (healthy disease) and males (healthy disease) might be a reason.

# Conclusion

In our study, we provide detailed information on proteomic profiles in cardiac remodeling because of severe AS and MR. This expands our knowledge about human cardiac remodeling in female and male patients with LV pressure and volume overload. In addition, the comprehensive data constitute a valuable basis for future analyses of cardiac function in human and preclinical research.

# Materials and Methods

## Patient cohort

41 patients with severe AS and 17 patients with moderate or severe MR (according to current diagnostic guidelines) were included in the study. Exclusion criteria were the presence of moderate-to-severe valve diseases of the remaining heart valves and general contraindications to CMR. Controls (n = 17) were 44 ± 15-yr-old healthy cardiac organ donors without cardiovascular diseases, whose hearts were not used for transplantation because of non-medical reasons.

The study protocol was in agreement with the principles outlined in the Declaration of Helsinki and was approved by the Medical Ethics Review Committee. All patients gave written informed consent before inclusion.

## Sample preparation for mass spectrometry measurements

### Left ventricular myocardial samples

Left ventricular myocardial samples were collected from patients with AS and MR during aortic or mitral valve replacement surgery or from healthy donor hearts not used for transplantation because of non-medical reasons. Samples were taken and frozen directly in liquid nitrogen and kept at −80°C. For protein extraction, biopsies were lysed in 200 μl lysis buffer containing the following: 2% SDS, 50 mM ammonium bicarbonate buffer, and EDTA-free Protease Inhibitor Cocktail (Complete, Roche). Samples were homogenized at room temperature using FastPrep-24 5G Homogenizer (MP Biomedicals) with 10 cycles of 20 s and 5-s pause between cycles. After heating the samples for 5 min at 95°C, five freeze–thaw cycles were applied. 25 U of Benzonase (Merck) was added to each sample, and after an incubation for 30 min, the lysates were clarified by centrifugation at 16,000*g* for 40 min at 4°C. Briefly, each sample was reduced with dithiothreitol (10 mM final; Sigma-Aldrich) for 30 min, followed by alkylation with chloroacetamide (40 mM final; Sigma-Aldrich) for 45 min and quenching with dithiothreitol (20 mM final; Sigma-Aldrich). Beads (1 mg) and acetonitrile (70% final concentration) were added to each sample, and after 20 min of incubation on an overhead rotor, the bead-bound proteins were washed with 70% ethanol and 100% acetonitrile. 2 μg sequence-grade trypsin (Promega) and 2 μg lysyl endopeptidase (LysC) (Wako) in 50 mM Hepes (pH 8) were added, and after an overnight incubation at 37°C, peptides were collected, acidified with trifluoroacetic acid, and cleaned up using the StageTips protocol.

Protein concentration was measured using the Bio-Rad DC Protein Assay, and 100 μg of each sample was further processed using the SP3 clean-up and digestion protocol as previously described (Hughes et al, 2019).

## Heart reference sample for matching library

A peptide mix for each experimental group (CON, AS, and MR) was generated by collecting 10 μg peptides from each individual sample belonging to the corresponding group. Equal peptide amounts from each group mixture were combined, desalted using a C18 SepPak column (100 mg; Waters), and dried down using a SpeedVac instrument. Peptides were reconstituted in 20 mM ammonium

formate (pH 10) and 2% acetonitrile, loaded on a XBridge C18 4.6 × 250 mm column (3.5 μm bead size; Waters), and separated on an Agilent 1290 HPLC instrument by basic reversed-phase chromatography, using a 90-min gradient with a flow rate of 1 ml/min, starting with solvent A (2% acetonitrile and 5 mM ammonium formate, pH 10) followed by an increasing concentration of solvent B (90% acetonitrile and 5 mM ammonium formate, pH 10). The 96 fractions were collected and concatenated by pooling equal interval fractions. The final 26 fractions were dried down and resuspended in 3% acetonitrile/0.1% formic acid for LC–MS/MS analyses.

### LC–MS/MS analyses

Peptide samples were eluted from stage tips (80% acetonitrile and 0.1% formic acid), and after evaporating, organic solvent peptides were resolved in sample buffer (3% acetonitrile/0.1% formic acid). Peptide separation was performed on a 20-cm reversed-phase column (75 μm inner diameter, packed with ReproSil-Pur C18-AQ; 1.9 μm, Dr. Maisch GmbH) using a 200-min gradient with a 250 nl/min flow rate of increasing buffer B concentration (from 2 to 60%) on a HPLC system (Thermo Fisher Scientific). Peptides were measured on an Orbitrap Fusion (individual samples) and a Q Exactive HF-X Orbitrap (reference sample) instrument (Thermo Fisher Scientific). On the Orbitrap Fusion instrument, peptide precursor survey scans were performed at 120K resolution with a $2 \times 10^5$ ion count target. $MS^2$ scans were performed by isolation at 1.6 m/z with the quadrupole, HCD fragmentation with normalized collision energy of 32, and rapid scan analysis in the ion trap. The $MS^2$ ion count target was set to $2 \times 10^3$, and the max injection time was 300 ms. The instrument was operated in top speed mode with 3-s cycle time, meaning the instrument would continuously perform $MS^2$ scans until the list of non-excluded precursors diminishes to zero or 3 s. On the Q Exactive HF-X Orbitrap instrument, full scans were performed at 60K resolution using $3 \times 10^6$ ion count target and a maximum injection time of 10 ms as settings. $MS^2$ scans were acquired in Top 20 mode at 15K resolution with $1 \times 10^5$ ion count target, 1.6 m/z isolation window, and maximum injection time of 22 ms as settings. Each sample was measured twice, and these two technical replicates were combined in subsequent data analyses.

### RAW data processing

Data were analyzed using MaxQuant software package (v1.6.2.6). The internal Andromeda search engine was used to search $MS^2$ spectra against a decoy human UniProt database (HUMAN.2019-01, with isoform annotations) containing forward and reverse sequences. The search included variable modifications of oxidation (M), N-terminal acetylation, deamination (N and Q), and fixed modification of carbamidomethyl-cysteine. Minimal peptide length was set to six amino acids, and a maximum of three missed cleavages was allowed. The FDR (false discovery rate) was set at 1% for peptide and protein identifications. Unique and razor peptides were considered for quantification. Retention times were recalibrated based on the built-in non-linear time-rescaling algorithm. $MS^2$ identifications were transferred between runs with the "Match between runs" option, in which the maximal retention time window was set to 0.7 min. The integrated LFQ quantitation algorithm was applied. LFQ values are given in Table S6. Gene symbols assigned by

MaxQuant were substituted with gene symbols of the reported UniProt IDs from the FASTA file used.

### Visualization

Schematic drawings were created using BioRender software. Heatmaps are drawn using the pheatmap R package (version 1.0.12). Proteins included in heatmaps were combined from gene names enriched in GO terms within a category. Condition group means of $\log_2$ (LFQ intensities) are centered, scaled protein wise, and clustered with default values.

All other plots were created using ggplot2, ggpubr, and cowplot R packages.

### Clinical imaging—cardiovascular magnetic resonance imaging and post-processing

#### *Left ventricular mass, volume, and function*
All CMR examinations were performed using a whole-body 1.5 T MR system (Achieva R 3.2.2.0; Philips Medical Systems) using a five-element cardiac phased-array coil. Analyses were performed using View Forum (View Forum R6.3V1L7 SP1; Philips Medical Systems Nederland B.V.). Gapless balanced Turbo Field Echo cine two-dimensional short-axis sequences were obtained using the standard CMR protocol for LV mass, volume, and function.

#### *Left ventricular fibrous tissue content*
For fibrosis assessment, a single breath-hold modified Look-Locker inversion recovery sequence in midventricular short-axis view was acquired before and 10 min after contrast administration. Calculation of extracellular volume (ECV) was performed using the following method:

$$ECV = (1 - hematocrit) * \frac{(1/T\ myo\ post) - (1/T\ myo\ pre)}{(1/T\ blood\ post) - (1/T\ blood\ pre)},$$

where myo = LV midwall myocardial T1 value, blood = LV blood pool T1 value, and pre and post refer to the measurement before and after contrast administration. Myocardial fibrous tissue content (absolute ECV = aECV) was calculated using the following equation: aECV = LV myocardial volume*ECV. LV myocardial volume = LV mass/1.05, where 1.05 is the myocardial density given in g/ml.

### Statistical analyses

Statistical analyses were performed using R (R version 3.5.3 and 4.0.3). MaxQuant results were filtered to exclude reverse database hits, potential contaminants, and proteins only identified by site, that is, proteins identified only by modified peptides. Furthermore, all proteins whose lead entry was marked "Fragment" or with <50% valid values in at least one compared group were excluded. LFQ values were $\log_2$-transformed, and missing values were imputed by random draw from the Gaussian distribution with 0.3*SD and downshift of 1.8*SD of the observed values per sample.

Linear models for the full sample set were calculated using the empirical Bayes procedures for residual variance estimation and mean–variance trend correction from limma (v3.38.3). Contrasts to

retrieve differential abundance were stated as follows: AS versus CON, MR versus CON, and AS versus MR for condition comparisons; and AS male versus CON male, AS female versus CON female, MR male versus CON male, and MR female versus CON female for sex-stratified comparisons. *P*-values are multiple testing corrected by the Benjamini–Hochberg methodology. Condition-specific effects are effects that are significant in a condition versus control and versus the respective other condition, whereas the direction, that is, negative (down) or positive (up) fold change, needs to be conserved. The effect in the other condition versus control has to be non-significant. Shared effects are defined through the same direction of significant effect in both conditions when they are compared with controls. Diverging effects show significant effects with opposing direction when compared with control samples. In the sex-stratified analysis, within a condition, we investigated proteins found significant in one sex only while assuming that significant changes found in both are also represented in the condition comparison.

Enrichment analysis was performed by the gprofiler2 R package (version 0.1.8) with a background set of all detected proteins and two query sets of all up/down-regulated proteins, respectively, per condition comparison and per sex-stratified comparison in order of largest to smallest absolute fold change against GO biological process, cellular compartment, and molecular function. *P*-values are controlled by FDR of 5%, and significant terms are filtered for sets in which the intersection size was <5% of the measured proteins of a set or less than one gene in the intersection. Gene sets needed a minimum size of three. Multiple entries with identical matched gene lists within a GO branch are reduced to the one with the lowest *P*-value. Further reduction in terms for pie charts was achieved via REVIGO using the following settings: medium reduction, against the *Homo sapiens* database, SimRel similarity measure, and without order of terms (Frangogiannis, 2019). For pie charts, we chose the most frequent representative of a cluster of redundant terms and calculated the proportions of frequency based on the assigned categories for the merged set of enrichments from up- and down-regulated proteins within disease groups.

Manual category assignments for GO terms are given in Tables S1 and S2. Organelle assignments are adopted from Doll et al by mapping UniProt protein IDs (Doll et al, 2017) (Fig S14). Enrichments belonging to the category "other terms" are based on 88 proteins with higher abundance in the diseased groups. Of these, 84% are typical body fluid components. Considering the different biopsy collection procedures for the sample groups, blood contamination becomes the most probable source of the signal and impedes any interpretation with regard to physiological differences in humoral immune response between the diseased and the control group. Enrichment analysis of AS-specific, MR-specific, and shared significant effects was performed by the gprofiler2 R package (version 0.2.0) (concerning data in Table S3) with a background set of all detected proteins and two query sets of up/down-regulated proteins, respectively, against GO biological process, cellular compartment, and molecular function. *P*-values are controlled by FDR of 5%, and significant terms are filtered for sets in which the intersection size was <5% of the measured proteins of a set or less than one gene in the intersection. Gene sets needed a minimum size of three. Multiple entries with identical matched gene lists within a GO branch are reduced to the one with the lowest *P*-value.

For the clinical characteristics of our study population, we compared the differences with a two-tailed *t* test in case of normally distributed numerical values or with a chi-squared test in case of discrete categorical values, both from the R stats package, version 3.5.1.

## Data Availability

Mass spectrometry raw data for the deep human reference proteome analysis of mixed patient samples and MaxQuant protein output tables for the entire cohort are available via ProteomeXchange with identifier PXD023800.

## Supplementary Information

## Acknowledgements

We would like to thank Alireza Khasheei (clinical imaging) for technical assistance and Manuela Bauer for her support as a study nurse. This work was supported by the German Federal Ministry of Education and Research (grant numbers 031A427A, 031A427B, 031A427C, 031A427D, and 01ZZ1802H), the ERA-CVD (SICVALVES), the CRC1470, and the European Commission under the H2020 Program (Grant No. 689617, Brussels, Belgium). M Kelm is a participant in the Charité Digital Clinician Scientist Program funded by the Deutsche Forschungsgemeinschaft (DFG). M Schafstedde is a participant in the BIH-Charité Junior Digital Clinician Scientist Program funded by the Charité-Universitätsmedizin, Berlin, the Berlin Institute of Health. I Baczko was supported by the Ministry of Human Capacities, Hungary (20391-3/2018/FEKUSTRAT). This work was also supported by the DFG (German Research Foundation) Grant SFB-1470, project Z03 (to T Kuehne), and B05 (to P Mertins).

### Author Contributions

S Nordmeyer: conceptualization, data curation, formal analysis, validation, investigation, visualization, project administration, and writing—original draft, review, and editing.
M Kraus: conceptualization, data curation, software, formal analysis, validation, investigation, visualization, methodology, and writing—original draft, review, and editing.
M Ziehm: conceptualization, software, formal analysis, validation, and writing—original draft, review, and editing.
M Kirchner: conceptualization, formal analysis, validation, visualization, methodology, and writing—original draft, review, and editing.
M Schafstedde: data curation, investigation, project administration, and writing—review and editing.
M Kelm: data curation, formal analysis, and writing—review and editing.
S Niquet: data curation, formal analysis, and writing—review and editing.
MM Stephen: software, formal analysis, and writing—review and editing.

I Baczko: data curation, investigation, and writing—review and editing.

C Knosalla: data curation, funding acquisition, investigation, and writing—review and editing.

M-P Schapranow: resources, supervision, funding acquisition, and writing—review and editing.

G Dittmar: conceptualization, funding acquisition, and writing—review and editing.

M Gotthardt: supervision, validation, and writing—review and editing.

M Falcke: resources, funding acquisition, and writing—review and editing.

V Regitz-Zagrosek: conceptualization, supervision, funding acquisition, and writing—review and editing.

T Kuehne: conceptualization, resources, supervision, funding acquisition, project administration, and writing—review and editing.

P Mertins: conceptualization, resources, supervision, funding acquisition, project administration, and writing—review and editing.

## Conflict of Interest Statement

The authors declare that they have no conflict of interest.

## Clinical Trial Registration

This study was registered at www.clinicaltrials.gov NCT03172338, NCT04068740.

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
