## [Reviewer comments · Life Science Alliance]

Life Science Alliance

Disease- and sex-specific differences in patients with heart valve disease - a proteome study

Sarah Nordmeyer, Milena Kraus, Matthias Ziehm, Marieluise Kirchner, Marie Schafstedde, Marcus Kelm, Sylvia Niquet, Mariet Stephen, Istvan Baczko, Christoph Knosalla, Matthieu Schapranow, Gunnar Dittmar, Michael Gotthardt, Martin Falcke, Vera Regitz-Zagrosek, Tltus Kuehne, and Philipp Mertins

DOI: <https://doi.org/10.26508/lsa.202201411>

Corresponding author(s): Sarah Nordmeyer, Deutsches Herzzentrum Berlin

Review Timeline:

Submission Date:	2022-02-15
Editorial Decision:	2022-06-10
Revision Received:	2022-11-15
Editorial Decision:	2022-12-08
Revision Received:	2022-12-16
Accepted:	2022-12-19

Scientific Editor: Novella Guidi

Transaction Report:

June 10, 2022

Re: Life Science Alliance manuscript #LSA-2022-01411-T

Dr. Sarah Nordmeyer
Deutsches Herzzentrum Berlin (DHZB), Department of Congenital Heart Disease - Pediatric Cardiology, Berlin, Germany
GERMANY

Dear Dr. Nordmeyer,

Thank you for submitting your manuscript entitled "Disease- and sex-specific differences in patients with heart valve disease - a proteome study" to Life Science Alliance. The manuscript was assessed by expert reviewers, whose comments are appended to this letter. We invite you to submit a revised manuscript addressing the Reviewer comments.

Thank you for this interesting contribution to Life Science Alliance. We are looking forward to receiving your revised manuscript.

Sincerely,

B. MANUSCRIPT ORGANIZATION AND FORMATTING:

Reviewer #1 (Comments to the Authors (Required)):

In this article Nordmeyer and colleagues have performed detailed proteomic analysis of heart tissue from patients who were afflicted with aortic valve stenosis or mitral valve prolapse. They have identified molecules that are differentially expressed in the diseased samples. Additionally, they have identified sex-specific differences in protein expression. The experiments are well done, and the data is easy to interpret. I have just 2 suggestions to improve the manuscript.

1. In figure 3 add some representative protein molecules in the plots.
2. I am guessing that the purpose of this study is to identify biomarkers that could be used to predict AS or MR. This anticipation can be clearly stated in the abstract and introduction. Additionally, none of the differentially expressed proteins (ECM, metabolism etc) can be easily used as biomarkers in the plasma. I suggest that the authors analyze their data and include at least 1 figure/table with a list of secreted molecules. Please refer to PMID: 27247337 for some molecules that can be included, such as ADM, ANP, CRP, ANGPT2 etc.

Reviewer #3 (Comments to the Authors (Required)):

In this manuscript, Nordmeyer and co-authors aim to perform a comprehensive proteomics comparison of two types of heart valve disease patients (aortic valve stenosis - AS - and mitral valve regurgitation - MR), taking also into account gender differences and clinical information of the patients. The study is very interesting and has several positive aspects, such as the fact of using two different forms of the disease, the gender aspect, and the combination of molecular and clinical information. Nevertheless, in my opinion, the authors decide to follow a path that makes the manuscript difficult to follow, and more importantly, does not fully respond to the goals proposed by the authors. Thus, in my opinion, this work needs to be revised prior to publication. Another aspect that is lacking in this work is the validation of the results. During the work, the authors already show that some results for some specific proteins are in line with the literature, but I also suggest that the authors try to compare their results with results from other screenings, including transcriptomic screening (if available) in a broader way (for instance by comparing all the altered protein found in different studies).

There are some aspects that need to be altered in order to improve the quality of the manuscript, namely:

1. The abstract should contain the main findings of the article; however, it is mainly limited to the general observations such as the number of altered proteins as well as the generic GO analysis. In my opinion, this is far from addressing the goals of the paper. Thus, in my opinion, more than indicating the number of proteins altered in each condition/gender, the aspects that must be clearly indicated in the abstract (be the focus of the paper) are: 1) which are the common and divergent mechanism in AS and MR; 2) Which are exactly the gender-specific alterations - it leads to different pathways being altered or just different proteins; and, 3) how does this is correlated with the clinical phenotype (image-based information and functional information).
2. Regarding the methods section, in my opinion, at least all the information regarding statistical analyses should be in the main text.
3. Results:
 - a. Section - Patient cohort: Table 1 should include information regarding the controls and the respective statistical analysis. Also, add the statistical analysis regarding the gender balance in the case of males (it is also strange that no statistical difference was obtained for females considering that there are 4-times more female patients in the AS group compared to MR). Highly the results where it is observed differences between groups and discussed them in the disease contexts. Moreover, the author should not only compare the three groups used in this assay, they should also see if their experimental groups are good representatives of the disease distribution within the population. Considering that the correlation of the proteomics data with the clinical evaluation is also one of the objectives of the study, supplementary figure 1 and 2 should be included in the main manuscript. Again, the author should pay attention to the parameter where differences were observed and discussed these results in a disease context.
 - b. Section - Hear proteome coverage and tissue specific features: contrarily to the previous section, the information present in this section, except for the PCA analysis, is not that relevant (it is very technical) for this work. Thus, both Fig2a and 2b can be moved to supplementary data and the PCA analysis can be included in the next set of results (Fig 3 where the group comparison starts). In a similar way, the text can be reduced/simplified and added to the next section.
 - c. Section - Quantitative proteome comparison of disease and healthy samples: considering the aim of the article this is the most important section and it will be also the one used to select the GO highlighted in the remaining article. In my opinion, the authors should go deeper in this analysis. The authors start by presenting the overall results (number of altered proteins), and they also

identify the set of proteins that are unique to a given disease as well as, the proteins commonly altered between the diseases. They even identify with the common ones which are the subset that present the same tendency and the ones that have an opposite profile. This is very interesting information, but the authors don't use it to perform the GO analysis and thus identify the unique and common (divergent and convergent) mechanisms that are altered between the diseases. Instead of that, the authors decide to perform a very generic GO analysis which end up revealing 4 broad terms which were then analyzed deeper. However, in my opinion with this way to present the results, the authors turn the narrative more difficult to follow, being most of the time mainly expressing the alterations that are more relevant in AS (where these categories are more or less equally represented in contrast to what happen in MR which considering the results presented are mainly characterized by an alteration at the cytoskeleton), moreover in most of the times it also missed the information regarding gender information due to lack of representation.

Thus, in my opinion, the way the authors decide to present the data not only turns the narrative more difficult to follow, as it also doesn't fully fulfill the aims of this work. Thus, I suggest that the authors alter the data analysis in a way that, after identifying the different sets of interesting proteins [(1) the unique proteins of each group; (2) the ones that are common to both groups (divergent and convergent profiles); (3) the gender-specific alterations (unique and common)] a GO enrichment analysis should be performed for each set of interacting proteins in order to identify the mechanisms that are associated with those alterations, and thus end up with a more comprehensive understanding of the differences and similarities between the two types of heart valve disease as well as how gender may have a different role in the disease.

Aspects that need to be discussed:

1. AS group is the one that presents more alteration, but it is also the group that has more individuals per group (almost double the other two groups). To which extent can this difference in terms of the number of individuals per group influence the results obtained? The author should elaborate on that.
2. There is a massive age difference between the three groups, with the AS group the one with a higher difference (mean difference of 9 years to MR and more than 20 years to CON). Again, the authors should comment on how thus this difference may influence the results obtained. If possible, identify other studies demonstrating that there are no differences in heart proteome with aging.
3. There is a massive difference between the number of proteins altered when a gender balance analysis is done compared to the one with all the subjects (usually there is twice the number of altered proteins between the two analyses). The authors should comment on that, including the hypothesis that this massive difference is due to the fact that there is also a reduction in terms of the number of individuals used to perform the analysis in the gender-balanced comparisons.
4. Considering that the authors have both moderate and severe MR, in my opinion, the authors should try to highly the differences associated with the severity and comment on how they think that these moderate cases can be responsible for less variation between the con and MR. On the other hand, the authors should also comment on the fact that besides having these two disease states the MR group presents less intra-group variability than the AS group, which is composed of severe cases only. There is any clinical justification for the larger variability of AS cases when compared to the MR cases, or this is in fact due to the difference in terms of the number of individuals per group? All these aspects may be discussed.

Minor aspects that need to be considered:

- Avoid abbreviations in the abstract and even if a given abbreviation was firstly indicated in the abstract, it should be also indicated in its extended form the first time it is used in the main document.
- Do a careful revision of the document, there are elements missing in some figures (which are referred to in the main manuscript). Do not use a single legend to indicate all the colors and different elements within the figures. Please, do it for every single element.
- Take into consideration the order by which each element is presented in the main manuscript and organized them within the figure in accordance with it, avoiding "jumping" within elements. It is easier to follow the article if the order is maintained. In the legend of the figures, please do not describe different elements that are distant from each other simultaneously. Describe each element separately and by order. E.g., "(C) + F) Comparison of fibrous tissue content as measured by MRI in AS and MR (C) and stratified to sex in AS (F)." In this example, start by describing only the C, then the remaining letter (D, E), and then the F.

Reviewer #1 (Comments to the Authors (Required)):

In this article Nordmeyer and colleagues have performed detailed proteomic analysis of heart tissue from patients who were afflicted with aortic valve stenosis or mitral valve prolapse. They have identified molecules that are differentially expressed in the diseased samples. Additionally, they have identified sex-specific differences in protein expression. The experiments are well done, and the data is easy to interpret. I have just 2 suggestions to improve the manuscript.

Comment 1:

In figure 3 add some representative protein molecules in the plots.

Answer 1:

Thank you very much for this comment, we added representative protein molecules in the plots in Figure 3 (new Figure 5).

Comment 2:

I am guessing that the purpose of this study is to identify biomarkers that could be used to predict AS or MR. This anticipation can be clearly stated in the abstract and introduction. Additionally, none of the differentially expressed proteins (ECM, metabolism etc) can be easily used as biomarkers in the plasma. I suggest that the authors analyze their data and include at least 1 figure/table with a list of secreted molecules. Please refer to PMID: 27247337 for some molecules that can be included, such as ADM, ANP, CRP, ANGPT2 etc.

Answer 2:

Thank you very much for this comment. Although the interest and demand in biomarker identification is high and proteomics technology represents a promising approach, it was not the goal of the present study. We focused on the systemic characterization of AS- and MR-associated disease alterations in the heart tissue proteome (covering mostly cellular and structural features) and developed study design and sample collection protocol accordingly. The aim was to gain molecular insight into similarities and differences in protein expression levels between AS and MR, thus between pressure overload (AS) and volume overload (MR) and between sexes, in order to better understand disease mechanisms. Only matched liquid biopsies (blood) would have allowed to draw meaningful conclusions related to liquid biomarker identification. And additionally, if having tissue proteomics and liquid biopsies we would need longitudinal follow up with different outcomes, in order to relate tissue and liquid biopsy findings to outcome. The mechanistic insight we try to detect by using proteomics is not primary relevant for disease diagnostics, but might help to improve and develop therapeutic approaches for heart valve diseases, with respect to disease type and sex.

Nevertheless, we followed the reviewers' suggestion and analyzed our data for secreted molecules based on the suggested publication. We found 37 proteins quantified and included in our group comparisons, 17 of which were significantly different between either AS, MR or both diseases compared to control, see heatmap below. However, in light of the difficulty in correctly interpreting abundance levels of the secretable but also in intra-cellularly occurring proteins from tissue samples, and the risk of misleading interpretations, we chose not include this figure in the manuscript or Supplement. All the information is however, available to the interested reader through the protein abundance values provided in Supplement Table 6 (former Supplement Table 5) (and supplement Table 1 for the significance annotations).

Reviewer #3 (Comments to the Authors (Required)):

Comment:

In this manuscript, Nordmeyer and co-authors aim to perform a comprehensive proteomics comparison of two types of heart valve disease patients (aortic valve stenosis - AS - and mitral valve regurgitation - MR), taking also into account gender differences and clinical information of the patients. The study is very interesting and has several positive aspects, such as the fact of using two different forms of the disease, the gender aspect, and the combination of molecular and clinical information. Nevertheless, in my opinion, the authors decide to follow a path that makes the manuscript difficult to follow, and more importantly, does not fully respond to the goals proposed by the authors. Thus, in my opinion, this work needs to be revised prior to publication. Another aspect that is lacking in this work is the validation of the results. During the work, the authors already show that some results for some specific proteins are in line with the literature, but I also suggest that the authors try to compare their results with results from other screenings, including transcriptomic screening (if available) in a broader way (for instance by comparing all the altered protein found in different studies).

Answer:

Thank you very much for these comments. As suggested by the reviewer we have searched in the literature for comparable datasets and found one very recently published study in which LV samples of patients with aortic valve stenosis and different degrees of heart dysfunction were studied and compared to healthy heart tissue by proteomic analysis (Brandenburg et al, 2022; J Mol Cell Cardiol.; <https://pubmed.ncbi.nlm.nih.gov/36084744/>). The results of a comparison to our data highlight great reproducibility between both studies and were included as a new Supplement Figure 2 and the following text section on page 6, lines 4-8 in the manuscript:

"Comparison of our results with a more focused LV proteomic study on AS subtypes with different disease burdens (Brandenburg et al) showed excellent agreement in the overlap of quantified proteins in general (Supplement Figure 2A) and also for AS- but less for MR- regulated proteins (Supplement Figure 2B,C). Higher numbers of significantly regulated proteins in our study can be explained by the deeper proteomic coverage and also overall larger sample size."

There are some aspects that need to be altered in order to improve the quality of the manuscript, namely:

Comment 1:

The abstract should contain the main findings of the article; however, it is mainly limited to the general observations such as the number of altered proteins as well as the generic GO analysis. In my opinion, this is far from addressing the goals of the paper. Thus, in my opinion, more than indicating the number of proteins altered in each condition/gender, the aspects that must be clearly indicated in the abstract (be the focus of the paper) are: 1) which are the common and divergent mechanism in AS and MR; 2) Which are exactly the gender-specific alterations – it leads to different pathways being altered or just different proteins; and, 3) how does this is correlated with the clinical phenotype (image-based information and functional information).

Answer 1:

Thank you very much for these valuable suggestions. We changed the abstract accordingly and hope the results and conclusion of this study are now better communicated.

Comment 2:

Regarding the methods section, in my opinion, at least all the information regarding statistical analyses should be in the main text.

Answer 2:

We moved all information regarding statistical analyses to the main text.

Comment 3a

Patient cohort: Table 1 should include information regarding the controls and the respective statistical analysis. Also, add the statistical analysis regarding the gender balance in the case of males (it is also strange that no statistical difference was obtained for females considering that there are 4-times more female patients in the AS group compared to MR). Highly the results where it is observed differences between groups and discussed them in the disease contexts. Moreover, the author should not only compare the three groups used in this assay, they should also see if their experimental groups are good representatives of the disease distribution within the population. Considering that the correlation of the proteomics data with the clinical evaluation is also one of the objectives of the study,

supplementary figure 1 and 2 should be included in the main manuscript. Again, the author should pay attention to the parameter where differences were observed and discussed these results in a disease context.

Answer 3a:

We included the sparse information available concerning the controls into the legend of Table 1. Regarding the gender balance we changed the phrasing in the Table. The statistical analysis describes that the gender balance was not significantly different between AS and MR; there were 4 times more female patients in the AS group, however, percentage-wise 51% of AS patients were female and 29% of MR patients were female, which might explain the lack of statistical significance in gender balance between AS and MR.

Patients with AS suffer from left ventricular pressure overload due to the increased gradient across the diseased aortic valve (Mean pressure gradient aortic valve, mmHg). Patients with MR suffer from mitral valve regurgitation (Mitral valve regurgitation, grade (none/mild, moderate, severe)), which leads to increase in left ventricular end-diastolic volume (Left ventricular end-diastolic volume, ml/m²). Since these parameters are disease specific for AS and/or MR we see the significant differences between groups, which highlights the fact, that these patients are good representatives of their respective disease group.

We included the following sentences into the Discussion (page 13 line 22-27):

“AS and MR patients in this study are good representatives for their disease. AS patients suffer from left ventricular pressure overload due to the increased gradient across the diseased aortic valve and patients with MR suffer from volume overload due to mitral valve regurgitation, which leads to increase in left ventricular end-diastolic volumes. These parameters are significantly different between AS and MR patients, while other potentially relevant clinical parameters, such as incidence of diabetes or type of medication, were not different between groups.”

Former Supplement Figure 1 and 2 were included into the main manuscript as new Figures 2 and 3. New Figure 2 visualizes the cardiac parameters also described in Table 1. New Figure 3 describes the sex differences in imaging parameters within the AS group and within the MR group.

Comment 3b:

Heart proteome coverage and tissue specific features: contrarily to the previous section, the information present in this section, except for the PCA analysis, is not that relevant (it is very technical) for this work. Thus, both Fig2a and 2b can be moved to supplementary data and the PCA analysis can be included in the next set of results (Fig 3 where the group comparison starts). In a similar way, the text can be reduced/simplified and added to the next section.

Answer 3b:

Thank you very much for these valuable suggestions. We moved the coverage information to the Supplement (Supplement Figure 1A and B) and kept the PCA plot in the manuscript.

Comment 3c.

Quantitative proteome comparison of disease and healthy samples: considering the aim of the article this is the most important section and it will be also the one used to select the GO highlighted in the remaining article. In my opinion, the authors should go deeper in this analysis. The authors start by presenting the overall results (number of altered proteins), and they also identify the set of proteins that are unique to a given disease as well as, the proteins commonly altered between the diseases. They even identify with the common ones which are the subset that present the same tendency and

the ones that have an opposite profile. This is very interesting information, but the authors don't use it to perform the GO analysis and thus identify the unique and common (divergent and convergent) mechanism that are altered between the diseases. Instead of that, the authors decide to perform a very generic GO analysis which end up revealing 4 broad terms which were then analyzed deeper. However, in my opinion with this way to present the results, the authors turn the narrative more difficult to follow, being most of the time mainly expressing the alterations that are more relevant in AS (where these categories are more or less equally represented in contrasts to what happen in MR which considering the results presented are mainly characterized by an alteration at the cytoskeleton), moreover in most of the times it also missed the information regarding gender information due to lack of representation.

Thus, in my opinion, the way the authors decide to present the data not only turns the narrative more difficult to follow, as it also doesn't fully fulfill the aims of this work. Thus, I suggest that the authors alter the data analysis in a way that, after identifying the different sets of interesting proteins [(1) the unique proteins of each group; (2) the ones that are common to both groups (divergent and convergent profiles); (3) the gender-specific alterations (unique and common)] a GO enrichment analysis should be performed for each set of interacting proteins in order to identify the mechanisms that are associated with those alterations, and thus end up with a more comprehensive understanding of the differences and similarities between the two types of heart valve disease as well as how gender may have a different role in the disease.

Answer 3c:

Following the reviewer's suggestion we performed additional GO enrichment analyses of the AS-specific, MR-specific and shared proteins with significantly higher or lower abundance, respectively. We included now the full results of these 6 enrichment sets in an additional Supplement Table 4. Previously, the enrichments of specific only and shared only effect characteristics were not provided in a formal way, but only in the description of the manuscript. With the addition of the new Supplement Table also the formal characteristics are now available. The results in these tables are in line with our previous findings of our study as described in more detail below.

Concerning the extracellular matrix (ECM), for example, we find e.g. "collagen-containing extracellular matrix" and "glycosaminoglycan binding" enriched in proteins increased in both diseases (shared up), while also finding significant enrichment of "collagen-containing extracellular matrix", "supramolecular fiber" and generally "extracellular matrix structural constituent" in AS-specific increased proteins (see lines "In general, ECM related proteins were significantly higher in abundance in AS and MR compared to controls (Figure 6A). [...] AS samples show specific enrichment of distinct collagen-related GO terms (Fig 6D). There is a pronounced higher amount of fibrillar collagens like collagen type I". This example already highlights, that separate descriptions/interpretations of the intersection and disease-specific groups could be misleading when scattered across multiple paragraphs and that the manuscript is easier to follow when structured via biological/GO categories.

As in the case of the enrichments of non-intersected differences, large number of significantly enriched terms e.g. 260 GO terms for the shared up, are found, whose structuring in broader biological terms such as ECM or energy metabolism and mitochondria is still extremely helpful in understanding and presenting these results as the comparisons can become quite complex. As such we have chosen to keep the original approach of the manuscript, presenting the characteristics of the disease in broad terms, within which more detailed groups and individual proteins are described highlighting shared and specific effects on each level, while now also referring to the suggested additional analysis results in the new supplemental table.

For the benefit of the reviewer we have added here a short description of each of the broad terms from the intersection point of view:

In addition to the above described ECM effects, we also find clearly signs of changed energy metabolism and mitochondria processes with "mitochondrion", "generation of precursor metabolites

and energy” and “aerobic respiration” being enriched in proteins with shared lower abundance (shared do). Stronger and or additional effects are evident from separate enrichment of “mitochondrion” in the AS specifically lower proteins. In the intersection enrichments we found only in AS specific lower abundance proteins terms associated with proteostasis such as “ribosome”, “translation”, “chaperonin-containing T-complex” as already described in the manuscript (page 10, line 1ff “We found downregulation of proteins belonging to GO terms describing translation including ribosomes, protein folding and quality control, i.e. chaperonin containing T-complex protein Ring Complex (TriC)”). Further in line with our previous manuscript description, we observe enrichments of cytoskeletal and contractile proteins (e.g. “actin binding”, “actin cytoskeleton” in shared up proteins). The complex nature of these changes described in the manuscript is showcased by the fact that “actin filament binding” is significantly enriched in MR specific lower abundance proteins (MR do) while “actin cytoskeleton”, “lamin binding” are enriched in MR specific more abundant proteins (MR up). The MR specific changes of “cytoskeletal proteins just beneath the sarcolemma” with higher in abundance of “proteins such as SPTBN1, SPTB” (manuscript text) are evident in the enriched term “spectrin” in MR specific up.

We have included the following text section on page 6 lines 15-18:

“GO enrichment analysis of up- and down regulated AS- and MR-specific or commonly regulated proteins also confirmed that the majority of proteome alterations in our study can be systematically grouped into the four categories described above (Supplement Table 4).”

Aspects that need to be discussed:

Comment 1.

AS group is the one that presents more alteration, but it is also the group that has more individuals per group (almost double the other two groups). To which extent can this difference in terms of the number of individuals per group influence the results obtained? The author should elaborate on that.

Answer 1:

Thank you very much for this comment. We performed additional down-sampling analysis to study possible influence of the number of individuals per group. For this we randomly selected 17 samples from the AS cohort and compared them to all 17 control samples and counted total differentially abundant proteins. This was repeated 100 times and the average number of differentially abundant proteins is 746 and as such less than in our full analysis (>1300). However, still approximately twice as many as in the MR vs CON comparison where group sizes are identical (400 differentially expressed proteins). A two-sided t-test against 400 resulted in a p-value of 2.2×10^{-16} . We can thus conclude that the higher number is not an exclusive effect of higher power, instead there is much more regulation even when equal sample sizes are considered. To capture the true variance of the AS group we included all samples into the analysis.

We included information into the discussion (page 14, lines1-5) and into Supplement material.

“We performed an additional analyses studying group size dependent influence on number of differentially expressed proteins of the AS group, which showed most alterations in the AS group independent on group size (Supplement Figure 13). We can thus conclude that the higher number of altered proteins in the AS group is not an exclusive effect of higher power, since there is more regulation when equal sample sizes compared to MR are considered.”

Comment 2.

There is a massive age difference between the three groups, with the AS group the one with a higher difference (mean difference of 9 years to MR and more than 20 years to CON). Again, the authors

should comment on how thus this difference may influence the results obtained. If possible, identify other studies demonstrating that there are no differences in heart proteome with aging.

Answer 2:

The difference in age between AS and MR and especially between controls and patients is a limitation of the study, we included this into the manuscript, however, due to the very limited availability of left ventricular myocardial samples a healthy control group matching in age was not possible to achieve. The age of AS individuals spans from minimum of 41 years to maximum 81 years and thus covers a fairly large range. Therefore, we performed differential abundance analysis using a linear modelling strategy of abundance in relation to age within the condition. Here, we did not find any significant up- or down regulation of protein abundance. The same is true for a comparison of protein abundance to age in MR only (29 -79 age span).

Additionally, in AS and MR the age is homogeneous between sexes. As such, a sole impact of age in our comparison of conditions is unlikely as, for example, the proteostasis effects are strongest in female AS.

From human autopsies an increase of collagen content, for example, is reported between 20-25 year old to 67-87 year old and in 80 year old subjects an increase of collagen I and decrease of Collagen III was found. In our study, we also see an increase in extracellular matrix proteins, however, in AS patients we found a specific increase of collagen I and III, for example, which we did not find in MR patients, suggesting rather a disease specific than an age specific expression pattern.

We included this information into the Discussion page 14 lines 15-28 and page 15 lines 1-4.

Comment 3:

There is a massive difference between the number of proteins altered when a gender balance analysis is done compared to the one with all the subjects (usually there is twice the number of altered proteins between the two analyses). The authors should comment on that, including the hypothesis that this massive difference is due to the fact that there is also a reduction in terms of the number of individuals used to perform the analysis in the gender-balanced comparisons.

Answer 3:

Thank you very much for addressing this point. We added a clarifying sentence into the discussion page 17, lines 17-20. "In general, the gender balance analysis rendered fewer altered proteins than the one with all subjects. The smaller number of subjects in the gender balance analysis as well as differences in regulation between females (healthy-disease) and males (healthy-disease) might be a reason."

Comment 4:

Considering that the authors have both moderate and severe MR, in my opinion, the authors should try to highlight the differences associated with the severity and comment on how they think that these moderate cases can be responsible for less variation between the con and MR. On the other hand, the authors should also comment on the fact that besides having these two disease states the MR group presents less intra-group variability than the AS group, which is composed of severe cases only. There is any clinical justification for the larger variability of AS cases when compared to the MR cases, or this is in fact due to the difference in terms of the number of individuals per group? All these aspects may be discussed.

Answer 4:

Thank you very much for this thoughtful comment. The reviewer is correct, that we have patients with diagnosed moderate and patients with diagnosed severe mitral valve insufficiency within in mitral valve group. However, all these patients were diagnosed as having a relevant mitral valve insufficiency with cardiac enlargement and clinical symptoms of heart failure and, thus, were classified as having an indication for operation. From a clinically point of view we would not necessarily believe that the moderate MR group has less myocardial changes than the severe MR group, since they presented similarly in clinical appearance and cardiac morphology. Nevertheless, we performed a Principal Component Analysis including moderate and severe MR patients and found very similar abundance variation (Figure 4B).

For AS patients, although all patients presented with severe aortic valve stenosis, sex differences in cardiac remodeling are well described in the literature, which is not the case for patients with mitral valve regurgitation, which we believe to be the reason for larger variability in a cohort with similar gradients across the aortic valve, but differences in phenotype and differences in protein expression levels.

We included a PCA including moderate and severe MR patients in Figure 4B and the following sentence into the Results (page 5, lines 16-18) "Although the MR cohort covers patients with moderate as well as severe mitral valve regurgitation, no clear separation of these two echocardiographically classified disease groups was observed on the proteome level (Figure 4B)." and into the Discussion (page 14, lines 6-9) "Additionally, we want to mention that in the MR group there are patients with echocardiographically classified moderate and those with severe mitral valve regurgitation, which might have been a reason for more intra-group variability, however, PCA analysis showed very similar protein abundance variation (Figure 4B).".

Minor aspects that need to be considered:

Comment 1:

Avoid abbreviations in the abstract and even if a given abbreviation was firstly indicated in the abstract, it should be also indicated in its extended form the first time it is used in the main document.

Answer 1:

We avoided all abbreviations in the abstract.

Comment 2:

Do a careful revision of the document, there are elements missing in some figures (which are referred to in the main manuscript). Do not use a single legend to indicate all the colors and different elements within the figures. Please, do it for every single element.

Answer 2:

We carefully revised the document and figures.

Comment 3:

Take into consideration the order by which each element is presented in the main manuscript and organized them within the figure in accordance with it, avoiding "jumping" within elements. It is easier to follow the article if the order is maintained. In the legend of the figures, please do not describe

different elements that are distant from each other simultaneously. Describe each element separately and by order. E.g., "C) + F) Comparison of fibrous tissue content as measured by MRI in AS and MR (C) and stratified to sex in AS (F)." In this example, start by describing only the C, then the remaining letter (D, E), and then the F.

Answer 3:

We carefully revised the document and figures and performed changes accordingly.

December 8, 2022

RE: Life Science Alliance Manuscript #LSA-2022-01411-TR

Dr. Sarah Nordmeyer
Deutsches Herzzentrum Berlin
Augustenburger Platz 1
Berlin 13353
Germany

Dear Dr. Nordmeyer,

Thank you for submitting your revised manuscript entitled "Disease- and sex-specific differences in patients with heart valve disease - a proteome study". We would be happy to publish your paper in Life Science Alliance pending final revisions necessary to meet our formatting guidelines.

- please address Reviewer 3's remaining comments
- please incorporate your supplemental materials section into the main Materials & Methods section; we do not have a word limit in this section
- please use the [10 author names, et al.] format in your references (i.e. limit the author names to the first 10)
- please add your supplemental figure legends and your table legends to the main manuscript text
- please add a figure callout for Figure 5B and Figure 7G to your main manuscript text
- dataset PXD023800 should now be made publicly accessible

A. FINAL FILES:

B. MANUSCRIPT ORGANIZATION AND FORMATTING:

Sincerely,

Reviewer #3 (Comments to the Authors (Required)):

I would like to thank the authors for the effort made to address the reviewers' comments/suggestions. In general, the authors addressed all my comments, and I am satisfied with the current version of the manuscript.

However, some minor aspects need to be addressed to improve the manuscript:

1. In the Results section, in the subsection devoted to the patient's cohort, the authors should describe the major findings, highlighting the parameters where differences were observed.
2. On page 5, line 19, please change the sentence "(...) (Figure 4A) shows a clear separation of AS from MR and CON (...)" to something like "(...) (Figure 4A) already reveal some degree of separation between AS (...)". The separation is not that clear, there are several individuals from different groups mixed and besides that, the separation observed using the PC1 and PC2 is less than 20%.
3. Figures 2 and 3: the authors could be reorganized differently, starting by presenting the parameters with statistical significance. Moreover, in Figure 3 the author could add the information regarding the group (AS or MR) on top of the respective panel. Interestingly, there were no differences in the MR group, this should be indicated in the results section.
4. The volcano plot in Figure 5C should have different colors, different from the A and B since it represents the comparison between AS and MR, thus indicating that yellow is the one altered in AS and the green in MR doesn't make any sense in this plot. The authors could use blue and red for the down and up-regulated proteins.
5. There are a few typos in the text, which should be corrected.

December 19, 2022

RE: Life Science Alliance Manuscript #LSA-2022-01411-TRR

Dr. Sarah Nordmeyer
Deutsches Herzzentrum Berlin
Augustenburger Platz 1
Berlin 13353
Germany

Dear Dr. Nordmeyer,

Thank you for submitting your Research Article entitled "Disease- and sex-specific differences in patients with heart valve disease - a proteome study". It is a pleasure to let you know that your manuscript is now accepted for publication in Life Science Alliance. Congratulations on this interesting work.

DISTRIBUTION OF MATERIALS:

Again, congratulations on a very nice paper. I hope you found the review process to be constructive and are pleased with how the manuscript was handled editorially. We look forward to future exciting submissions from your lab.

Sincerely,
